# Planning Criteria and Models for the Development of Urban Structures of Coastal Settlements of Boka Kotorska

**Sanja Savić \*, Nevena Mašanović and Jelena Bajić Šestović** 

Faculty of Architecture, University of Montenegro, 81000 Podgorica, Montenegro; nevena.m@ucg.ac.me (N.M.); jelena.bajic@ucg.ac.me (J.B.Š.)
\* Correspondence: sanjav@ucg.ac.me; Tel.: +382-69-337865

**Abstract:** This study examines planning criteria and models for the development of urban settlements situated along the coastline of Boka Kotorska. In the period between 1940 and 2006, coastal settlements underwent great transformations that changed their historical image and the image of the immediate landscape significantly. The transformations resulted from political, economic and sociological changes occurring within the observed period. Consequently, huge transformations of the urban structure and the appearance of certain settlements are noticeable. Based on the available archival material (topographic maps, plans and drafts of the settlement or its parts in different scales), statistical data and the analysis of completed scientific research in this area, the analysis and comparison of the expansion of Boka Kotorska settlements as well as analysis of their urban transformation was conducted. Defining and using criteria based on recognizing the features of the subject area is a prerequisite for the planned development of Montenegrin coastal settlements, both regarding built and nonbuilt or ruined structures of the subject area. By identifying planning criteria and models, and upon conducting the systematic examination and valuation of physical planning documentation, new planning criteria and models for the improvement of Boka Kotorska coastal settlements have been proposed—ones that can be used in practice, i.e., for the development of physical planning documentation, studies and expertise.

**Keywords:** coastal settlements of Boka Kotorska; planning criteria and development models; built-up structures; urban transformation; physical planning documentation

## 1. Introduction

Major urban transformations of coastal settlements in Europe and the world have been identified in the last three decades. Through scientific research conducted in order to collect data for the assessment of urban changes in coastal settlements, estimates of the future state of these areas were given. Major changes were observed in land cover and population, agricultural and forest areas, natural areas, maritime economy and coastal use. Therefore, through studies and reports prepared on the topic of coastal settlements, an attempt is made to find a method of balancing the need for development and protection of the very resources that sustain coastal economies.

According to EEA Report No 6/2006 [1], three main types of coastal land use trends occurred between 1990 and 2000: artificial surfaces increased during the period, pasture and mixed farmland showed a major decrease, and arable lands and permanent crops increased. Studies were carried out on the definition of coastal setback and criteria derived from natural and physical characteristics and consequent vulnerability of the coastal zone [2] and programs were made on integrated coastal management of Montenegro [3], conducted on the basis of European reports [1]. A special factor of spatial development is the pressure of foreign and private capital which, being a constant of the capitalist era, is an inevitable factor in the further urban development of settlements [4]. The scarcity of public resources has generated numerous legislative provisions aimed at regulating the privatization of

positive externalities deriving from investments and interventions in the urban context [5]. The research conducted from the perspective of the changes of the built structures of the area in the observed period has shown that there is an increasing pressure on the construction area of coastal settlements of Montenegro, as well as in Europe, which triggers the need to preserve the identity of the area and the obligation to change the methodological approach to the development of physical planning documents (PPD).

In this way, it was undoubtedly confirmed that built-up land and its change through surface area and position in a certain period is the biggest indicator of changes in the area of coastal settlements. It was these insights that proved to be very important in the process of identifying the directions, and later monitoring, of the planned development of settlements in the area. In order to establish a harmonious relationship between settlements and the immediate landscape, this paper will attempt to determine general and special planning criteria for the development and improvement of coastal settlements and propose possible planning models for their future development. The paper is divided into three parts: introduction to the subject and methods of research, research results and concluding considerations.

In the first part of this paper, through the legislative literature and physical planning documentation, the concepts that will be used in the design and description of the selected spatial units will be analyzed, as well as other concepts of importance for determining the criteria for the creation, development and improvement of settlements. Throughout the research, the legal regulations from the period of 1945–2006 [6–14] were considered. Since the physical planning documentation was adopted pursuant to the said regulations, they represent an integral part of the subject research.

In the Results section, this study provides a comparison of data related to two time frames: for the period from 1945 to 1992, and for the period from 1992 to the independence of Montenegro (2006), with the aim of observing major changes in the urban structure and appearance of individual coastal settlements. The planned development of the subject area has was observed through the analysis of the situation after World War II through the physical planning documentation of the state and regional level [15–29], as well as through the physical planning documentation of the local level for two or more coastal municipalities [30–34]. A special emphasis has been put on planning documentation of the local level for municipalities of Kotor [35–41], Tivat [42–50] and Herceg Novi [51–54].

The Results section it is divided into five parts. In the first part, established identity attributes, i.e., changes in built-up land, were analyzed. The aim of the results obtained in this way is to determine the procedure for monitoring changes in space and time in the area of coastal settlements. The second part of Results section represents the determination of identity attributes that influenced the planned development of these settlements. In the third part of the Results section, this study determines the planning criteria for the development of coastal settlements, all based on previously recognized identity attributes. The procedure for determining planning models based on previously recognized identity attributes from physical planning documentation is presented in the fourth section.

Based on the reviewed existing physical planning documentation and the recognized and established planning criteria and models for the further development and improvement of coastal settlements, and in accordance with their natural and historical–cultural features, the last section will consider new planning criteria and guidelines for their future development.

By identifying the characteristics of the coastal settlements in the municipalities of Kotor, Tivat and Herceg Novi during the observed period, as well as the criteria and models for the improvement of spatial development thereof, concrete proposals and solutions for the planned development were established. The aim of proposing possible planning models that will include recommendations for future interventions is to create a new or supplement and preserve the existing identity of the settlements (characteristics of the origin and development), as well as to establish a harmonious relationship between the settlement and the immediate landscape.

In this research, the procedure for recognizing and determining changes in built-up land in the area of coastal settlements was given. The analysis of built-up land presented in this work is certainly a unique way of presenting the spatial changes of any coastal settlement. A certain scientific contribution was made in the field of research on the spatial development of settlements throughout history, regardless of the fact that in this research it was conducted only for the period from 1945 to 2006.

Planning criteria and guidelines are recognized through previously analyzed and evaluated physical planning documentation. The process of recognizing the impact of certain identity attributes of the area on the spatial development of coastal settlements and determining the criteria for their improvement and development represents the scientific contribution of this paper. Additionally, a certain scientific contribution was made in the field of research on the spatial development of settlements throughout history, regardless of the fact that in this research it was conducted only for the period from 1945 to 2006.

## 2. Materials and Methods

Population density and changes in built-up area are observed through European reports as the most significant factors of spatial changes. Within the 10 km coastal zone, urban surfaces are dominant on the first kilometer from the shoreline. Population densities are also higher on the coast than inland. For Europe, population densities of the coastal regions (NUTS3) are on average 10% higher than inland (Figure 1). However, in some countries, this figure can be more than 50% [1].

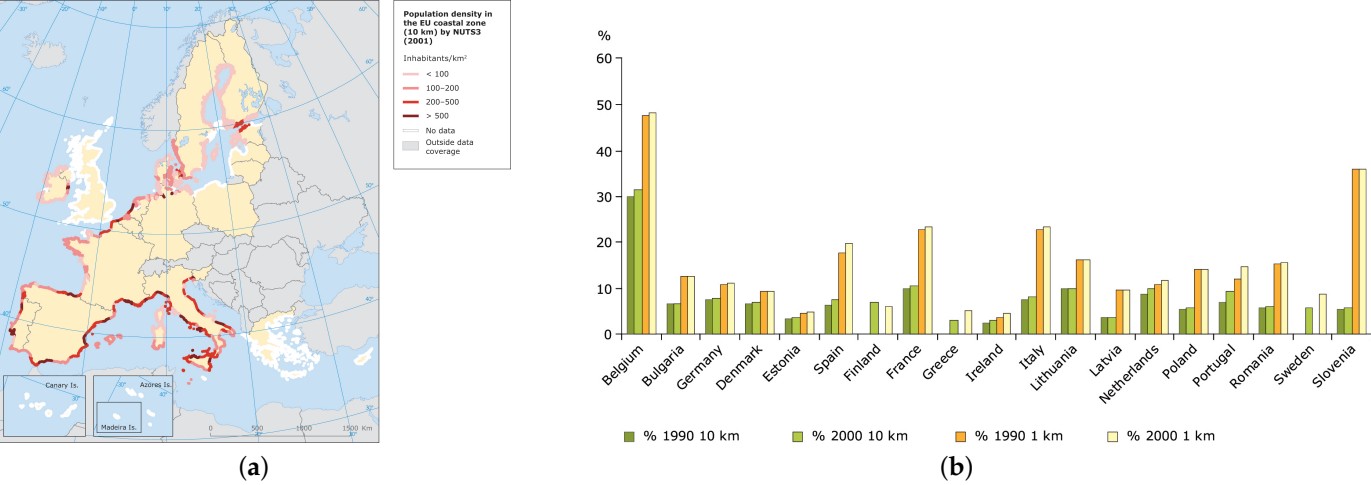

**(a)** **(b)**

**Figure 1.** (**a**) Population density in the European coastal zone (0–10 km) in 2001. (**b**) Built-up area in the 0–1 km coastal strip versus the entire 10 km coastal zone (1990–2000).

Therefore, this research will analyze the change in the number of inhabitants in coastal settlements through the coefficient of littoralization and its impact on spatial changes, i.e., increase/decrease in built-up area. The goal of this data analysis is to recognize their impact on the planned development of the subject area (Table 1).

Analysis and evaluation of the available material is grouped into two subchapters: definition of the subject of research, time frame and analysis of terms used in the research; systematization of Montenegrin legislation relevant to the development of physical planning documents; and systematization, analysis and evaluation of physical planning documentation of coastal settlements of Montenegro adopted in the observed time period.

**Table 1.** Methodology approach used in this research.

| Subject of Research | Used Documentation | Methods/ Procedures | Aim/Contribution of the Research |
|---|---|---|---|
| Conceptual definition of spatial units | → Legislative, by-law documents and physical planning documents | → Analysis, comparison and creation of cartograms | ⤏ Proposal for the introduction of new descriptions for existing concepts, as well as new concepts with detailed descriptions |
| Determination of time dividers | → Legislative documents, physical planning documents, constitutions, declarations | → Analysis and comparison of changes in the constitutional and political, legal and spatial planning framework of Montenegro | ⤏ Division of periods for easier monitoring of the most significant changes in the occurrence of coastal settlements and their results |
| The influence of state and local physical planning documents and legal acts on the planned development of coastal settlements | Legislative and by-law documents | Analysis of the existence of identity attributes in physical planning documents and criteria for their improvement | ⤏ Understanding the process of creating PPDs and their impact on the development of coastal settlements (procedure for determining criteria for analysis and evaluation of PPD) |
| Identity attributes of the coastal settlements of Montenegro | → PPD, state maps, statistical data | → Graphic overlay of data, analysis and comparison | ⤏ Determination and evaluation of identity attributes and their impact on changes in space (built-up land) |
| Determination of planning criteria for the development and improvement of coastal settlements | → PPD | → Analysis and comparison | ⤏ Grouping of identity attributes as a basis for determining planning criteria |

Proposal to supplement the planning criteria
(Determination of coefficients and capacity through mathematical equations based on identity attributes of coastal settlements)

| Subject of Research | Used Documentation | Methods/ Procedures | Aim/Contribution of the Research |
|---|---|---|---|
| Determination of general and special models for the development and improvement of coastal settlements | → PPD | → Analysis and comparison | ⤏ Grouping of identity attributes as a basis for determining planning models |

Proposal to supplement the planning models (based on the newly established planning criteria)

After defining the subject of the research and the time frame, this research will analyze the legislative, by-law documents and PPD, all with the aim of obtaining the following results:

- Clearer spatial determination of areas with similar identity attributes for which general or special criteria for their development and improvement could be used (coastal municipality and coastal settlement);
- Defining and describing the term for land that is not built in accordance with the PPD (built-up land), all in order to conduct an analysis and determine changes in the areas of coastal settlements.

After the introduction of new descriptions for existing terms, as well as new terms with detailed descriptions, this research will analyze legal documents from the observed period of importance for making the decision on the development and adoption of the spatial planning document. This process is necessary in order to understand the preparation and adoption of the PPD, and to determine those features of legal documents that have influenced the quality of the preparation of physical planning documentation. After the systematization and evaluation of the available PPD, an analysis and evaluation of the identity attributes of the natural and anthropogenic environment of the coastal settlements (cultivated, historical–cultural and built) recognized through the PPD will be performed. This analysis was carried out through a previously determined grouping of identity attributes: features from the natural and cultivated environment, and features from the built environment (network of settlements, division of urban functions within the

Coastal Region and population distribution-demographic characteristics, transport links of coastal settlements and economic characteristics).

The aim of this research is to identify their possible influence on the planned development of the settlement and on changes in their appearance. Through a detailed analysis and after observing certain vagueness or even the absence of some very important criteria, in the conclusion of this chapter, a newly proposed planning criterion (capacity and coefficient of the beach/coast) will be proposed, which is directly related to identity (natural and demographic) attributes.

### 2.1. The Subject of the Research, Time Frame and Terminological Determination

The analysis of previous research on coastal settlements in Montenegro has not been conducted in detail in legal and professional literature. Geographer Branko Radojičić compiled the systematization of settlements and regions in Montenegro based on the definition of Miloš Macura formed after World War II for the territory of the former Yugoslavia [55]. Ethnologist Petar Vlahović also researched the development of Montenegrin settlements and described the historical period of their formation, their appearance and changes depending on the period of their development [56]. Sociologist Vujović explored the changes that Montenegrin coastal settlements faced after World War II by addressing the transformation factors, yet not referring to their influence on the urban development [57]. Demographic characteristics of the coastal settlements of Montenegro from the middle of the 20th century were examined in the work of a geographer Pavle Radusinović, who compared and analyzed cause-and-effect relationships thereof [58]. Significantly greater contribution was made by a doctoral dissertation and subsequently published book written by tourismologist Devedžić Mirjana. In the said doctoral dissertation, she presented the typology of coastal cities as homogeneous, monofunctional and heterogeneous units of Montenegro, depending on the type of tourist motives, their location and traffic connectivity directions. In her work, she provided a systematic overview of three main criteria for settlement expansion, as well as the problems of the development of secondary residences [59,60]. Jurist Filip Turčinović researched the concepts of natural, cultural (immovable and movable) heritage of the Montenegrin coast through legal regulations and chronologically systematized charters, declarations, as well as international gatherings [61]. Even though there is no specific scientific research on spatial development and the improvement of coastal settlements in Montenegro from the urban perspective, the research conducted through physical planning documentation dated 1945–2006 should not be neglected, since professional methods can be part of scientific methodology used to research the model of planned development and improvement of a coastal settlement. Urban and architectural genesis of the origin and development of coastal settlements of the municipalities in question is described in studies that are part of the physical planning documentation [62–71]. Most coastal municipalities were developing spontaneously until World War II, even though the planning documents that testify to the suggestions for their further spatial development had also existed before the observed period [72].

The littoral or southern region of Montenegro covers the area of six administrative units [73], i.e., 6 municipalities and 250 settlements [74]. The subject of this research refers to the settlements located in the coastal municipalities of Kotor, Tivat and Herceg Novi, i.e., the settlements of the said municipalities that are located on the Montenegrin coast. Settlements that belong to the area of the respective municipalities of Montenegro, but are not located on the coast, are referred to in this research as "other settlements" of coastal municipalities (Figure 2).

The period after World War II, when the historical image of coastal settlements as well as the image of their immediate landscape transformed significantly, represents a long time frame, and thus, for easier analysis of development and improvement of coastal settlements as well for monitoring their results, year 2006 has been taken as the final year of the observed research period. The comparative analysis of time determinants used in this research defines two subchapters: the period 1945–1992 and the period 1992–2006.

The timelines have been determined on the basis of changes in the constitutional and political (state-legal), as well as the legal and spatial planning framework of Montenegro (Figure 3). Hence, the following years have been selected as the start and end years of timelines representing the aforementioned periods: the final year of World War II, the year of the change of the political and constitutional framework of the state, as well as the year of independence of the Republic of Montenegro.

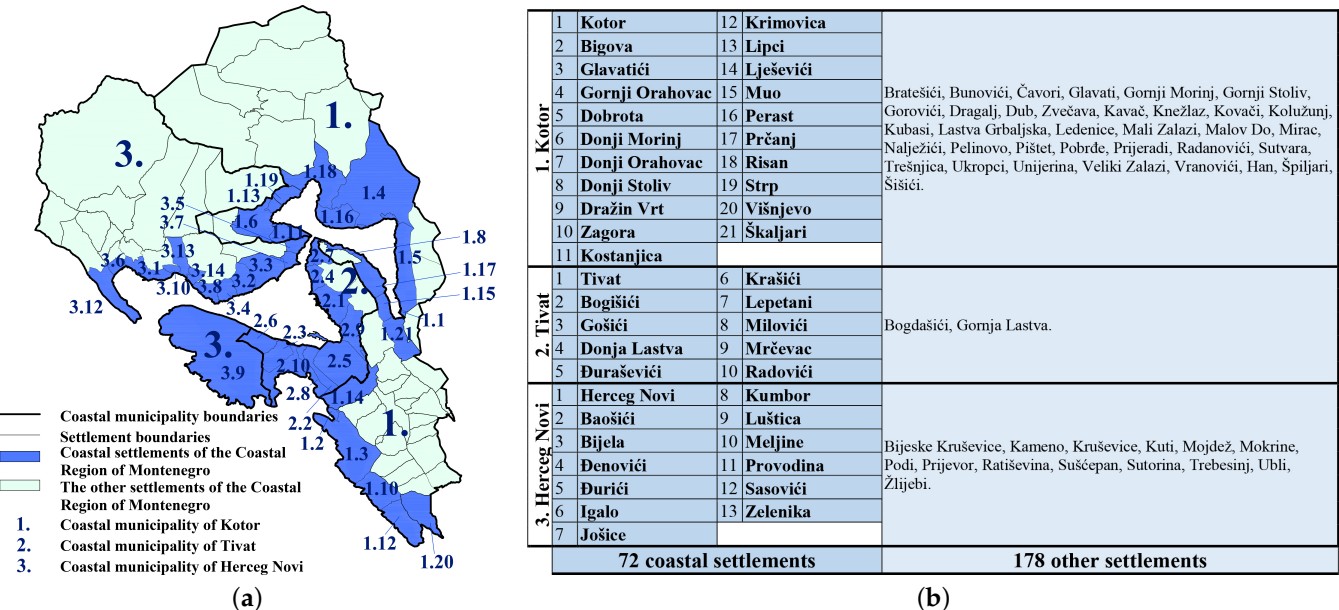

**Figure 2.** Settlements of coastal municipalities of Kotor, Tivat and Herceg Novi: (**a**) Position of coastal settlements and other settlements of the subject municipalities. (**b**) Names of coastal settlements and other settlements of the subject municipalities.

In the period after the independence of Montenegro (2006), no significant changes in the constitutional and political, legal and spatial planning framework of Montenegro were recognized in this research. Therefore, the year 2006 was taken as the final year for the analysis of spatial changes in the coastal settlements of Boka Kotorska.

In order to understand the terms coastal municipality and coastal settlement used in the legal, spatial and professional literature of Montenegro, the following terms are described in detail: coastline and seashore, construction area and built-up building land.

The terms "coastline" [75] and "sea shore" [28] are defined by the legislation of Montenegro, and the definitions provided therein are adopted in this paper. In the EEA Report No 6/2006 [1], the coastal zone is interpreted as the resulting environment from the coexistence of two margins: coastal land defined as the terrestrial edge of continents, and coastal waters defined as the littoral section of shelf seas. Legal documentation [14] provides definitions of the terms "construction area" and "building land". Pursuant to the current Law of Montenegro [14], the term "building land" refers to the land on which structures have been constructed and "land intended for regular utilization of structures, as well as land intended for building and utilization of structures in accordance with the planning document". Therefore, the term "construction area" "is closely linked with physical planning documentation, i.e., it represents its integral part. On the other hand, the term "building land" represents the current state of space and can be associated with the physical planning document only under the category of built-up building land, or the land on which structures have been constructed in accordance with the planning document. In further research, for the analysis and evaluation of physical planning documentation, the term "built-up building land" will be used for the land (cadastral parcels) on which the structures have been built (regardless of whether they are in accordance with the physical planning document).

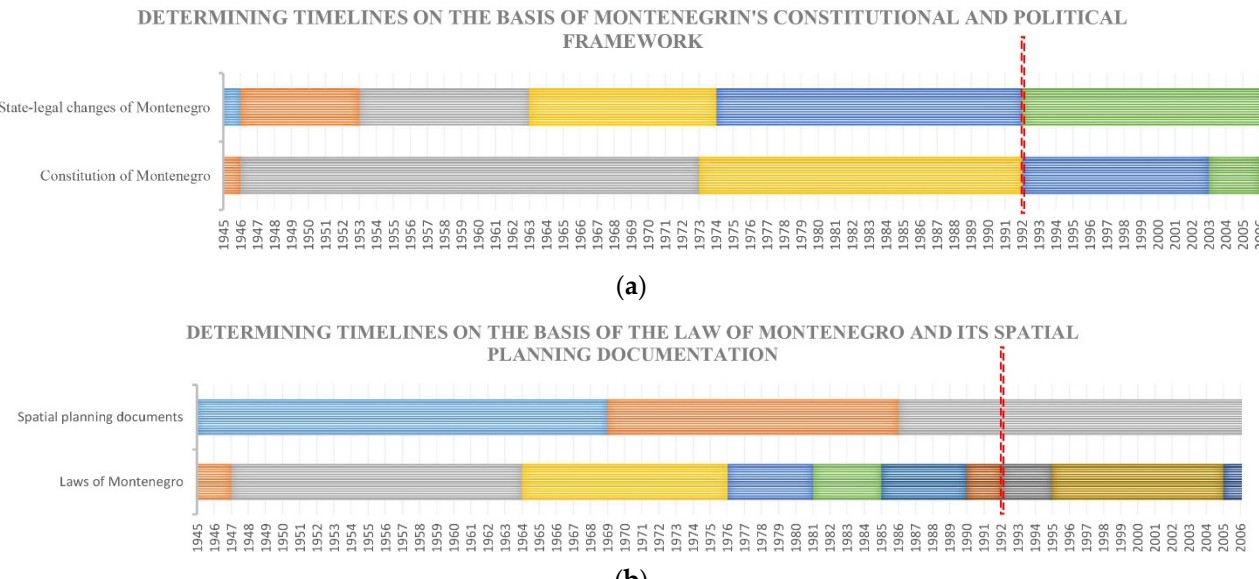

**Figure 3.** Time determinants set for the period 1945–2006: for state-legal changes and changes of the Constitution of Montenegro (**a**); for legal changes and changes of Montenegrin higher level physical planning documentation (**b**).

## 2.2. Planning Legislation as well as Analysis and Evaluation of Physical Planning Documentation of Boka Kotorska Adopted in the Observed Period

The Construction Act [76] was the first law that was valid in the observed period and was in force until 1964. Seven laws and their amendments [6–13] were adopted during the observed period and repealed by the adoption of legal documents after the observed period (after 2006), i.e., the adoption of the Law on Spatial Planning and Construction [77]. Table 2 shows the type of physical planning documents (marked with an "x" in the cell) according to seven legal documents adopted in the observed period.

Pursuant to the aforementioned laws, decisions on the adoption of physical planning documentation for the subject coastal settlements were issued. As of the higher-level plans dated after 1945, according to the 1964 Act [6], the subject area was covered by the Regional Spatial Plan: Montenegrin Littoral from 1966 [78] and Regional Spatial Plan: Southern Adriatic from 1968 [79]. Pursuant to the second Act adopted during the observed period [7], and its amendments [8], the decision on the adoption of the Spatial Plan of the Republic of Montenegro from 1986 [15] was issued.

**Table 2.** Physical planning documents compiled for the area of municipalities of Kotor (KO), Tivat (TV) and Herceg Novi (HN).

| Acronym of Law | LURP [6] | LSPA [7,8] | LSPA [9] | LSPA [10] | LSPA [11] | LSPA [12] | LSPA [13] | Total Number of SPD: |
|---|---|---|---|---|---|---|---|---|
| Title of the Type of Physical Planning Documents (Acronym) | 1964/1971 1973 | 1976/1979 | 1981 | 1985 | 1990 | 1995 | 2005 | |
| Program of the Regional Plan (PRP) | 1 (H) | | | | | | | 1 |
| General Regional Plan (GRP) | x (H) | | | | | | | - |
| Spatial Plan of the Republic (SPR) | | x (H) | x (H) | 1 (H) | 1 (H) | 1 (H) | - | 3 |
| Regional Spatial Plan (RSP) Spatial Plan of the Region (RSP) | 1 (H) | 1 (H) | x (H) | x (H) | x (H) | x (H) | | 2 |
| Spatial Plan of a Special Purpose Area (SPSPA) | | x (H) | x (H) | x (H) | x (H) | x (H) | x (H) | - |
| Municipal Spatial Plan (MSP) | | 1 (H-HN) | x (H) | 1 (H-KO) | x (H) | | | |
| Spatial Plan of the Local Self-Governance Unit (MSP) | | | | | | 1 (H-KO) | x (L) | 3 |
| General Urban Plan (GUP) | 1 (L-KO); 1 (L-TV); 1 (L-HN) | x (H) | x (H) | 1 (H-KO) | x (L) | 1 (L-KO) | x (L) | 5 |
| Detailed Spatial Plan (DSP) | | | | | x (H) | x (H) | x (H) | - |
| Local Study on Location (LSL) | | | | | | | x (L) | - |
| Study on Location (SL) | | | | | | | x (H) | - |
| Urban Program (UPg) | 1 (L-HN) | | | | | | | - |
| Smaller Settlement Spatial Plan (SSSP) | | x (L) | x (L) | x (L) | | | 2 (L-KO) | 2 |
| Settlement Spatial Plan (SSP) | | | | | x (L) | | | |
| Urban Settlement Plan (USP) | | | | | | x (L) | | |
| Detailed Urban Plan (DUP) | 2(L-KO); 1(L-TV); 1(L-HN) | 1(L-TV) | x (L) | x (L) | x (L) | x (L) | x (L) | 5 |
| Urban Project (UP) | | x (L) | x (L) | 4 (L-TV) | x (L) | x (L) | x (L) | 4 |
| **Total number of analyzed and evaluated physical planning documents related to the subject area, adopted (made) during the observed period:** | | | | | | | | **25** |

As of lower-level plans adopted pursuant to the aforementioned Act, the scope of which included the subject coastal settlements, the following documents were issued: UPg Herceg Novi from 1967 [53], GUP Boka Kotorska-Herceg Novi from 1970 [52], GUP Boka Kotorska-Tivat from 1970 [42] and GUP Boka Kotorska-Kotor from 1970 [80]. Based on the Detailed Urban Plans adopted according to the 1964 Act [6], which were available at the time of the research, the following were made: DUP Igalo from 1969 [54], DUP Markov Rt from 1970 [40], DUP Morinj from 1970 [39] and DUP Pržna from 1969 [50]. With the adoption of the 1976 Act [7], the conditions for the drafting of the 1979 MSP Herceg Novi [81] were met. Based on the amendments to this Act [8], the decision was made on the project consisting of the review or drafting of 31 plans, i.e., the review of SPR Montenegro, the drafting of eight MSPs including MSP Herceg Novi from 1979 [81] and the drafting of eighteen GUPs and four SSSPs. By amending the law from 1979, and after the natural disaster that struck the coastal region in 1979 [82], the development and review of another international project (in addition to the Southern Adriatic Project) was initiated with the participation of international institutions and organizations.

Even though the law (LURP and LSPA) [7,8] defined the term SPR in 1964, the first physical planning document of the highest level that was issued after World War II is Montenegrin SPR from 1986 [31], and it was adopted pursuant to the 1985 law [10]. Based on this planning document, the following were prepared: GUP Tivat from 1987, GUP Kotor from 1987 [35], MSP Kotor from 1987 [36], MSP Herceg Novi from 1989 and GUP Herceg Novi from 1989. The said law represented legal basis for deciding on DUP Tivat from 1980 [43–45].

Even though the 1990 law 1990 [11] revoked SSP that was established by the 1985 law [10], the draft of SSSP Prčanj [41] was made in 1992, as well as the drafts of SSSPs Stoliv and Prčanj [41]. While this law was in force, the amendments to SPR of Montenegro were

adopted in 1990 [16]. Pursuant to the law from 1995 [12], new amendments to the following were made: Spatial Plan of the Republic in 1997 [17], Spatial Urban Plan of Kotor from 1995 [37], and General Urban Plan of Kotor from 1998 [38]. Pursuant to the 2006 law [13], none state physical planning document, which was valid in the year when this study was conducted, was adopted, whereas four local physical planning documents related to the subject coastal area of Montenegro, which were still in force when the study was conducted, were adopted [46–49].

To sum up, the total number of physical planning documents that were made or adopted during the observed period (1945–2006) and include coastal settlements of Boka Kotorska partially or completely, and were available during the execution of this study, is 25 (Table 2). In this table, physical planning documents that were not made during the period of the law or were not available to the authors of this research are marked with "-", while the label "L" indicates the lower-level plans and label "H" stands for higher-level. The number in the table next to the "H" or "L" indicates the number of PPDs that were available to the authors of this study.

Through the analysis of the abovementioned documents, identity features were recognized, the improvement criteria of which are based on the planned development of the settlement. The criteria on which the development and improvement of coastal settlements are based depend on the location of the place in relation to the sea; traffic; the region; tourist settlements and their past significance; the quality and type of beaches; the quality of the landscape; natural conditions; spatial possibilities; past functions and their possibilities; development; existing construction; soil quality; cultural–historical monuments; as well as infrastructure development opportunities [83]. The criteria for the development and improvement of coastal settlements are grouped based on the similarity and connection of certain identity features and mutual influence. In their work [84], authors K. Petričić and M. Obad-Šćitaroci propose the following criteria for the evaluation of the settlement: natural features of the landscape and accommodation of the settlement (the criteria evaluate the typological features of the accommodation, geomorphological features and the uniqueness and preservation of the accommodation), historical significance (the criteria evaluate the age determination, historical functions, contents and placement in the space), spatial organization, type and structure of the settlement (criteria value the type of settlement, preservation of the original spatial organization and the impact of new construction on the traditional spatial organization), construction of the settlement (criteria value the preservation of traditional architecture, public/historical buildings and equipment as well as new construction and its impact on the settlement) and landscape-aesthetic features (criteria value the integration of the settlement into the landscape, visual exposure, panoramic views from the settlement, visual experience of the settlement and the importance of the location in relation to main roads, tourist routes, viewpoints, etc.). However, in this research, slightly different criteria for evaluating the PPD of coastal settlements are proposed, namely:

-   Criteria of natural values (based on identity attributes of the natural environment);
-   The criteria of cultural–historical values (based on the identity characteristics of the historical–cultural environment);
-   Spatial–urban criteria (based on the identity attributes of the built environment);
-   Economic criteria (based on economic identity characteristics).

Through the analysis and evaluation of PPDs, the following groups of identity characteristics were recognized, as well as criteria for the evaluation of coastal settlements (based on similarities and their mutual influence):

-   Features from the natural environment;
-   Features from the cultural and historical environment;
-   Features from the built environment;
-   Economic characteristics.

Therefore, in the continuation of this work, the features from the built environment (built-up area) of coastal settlements and other identity features recognized from the PPD

will be analyzed, all with the aim of recognizing the planning criteria and models of their development.

## 3. Results

By systematizing the abovementioned spatial planning documents as well as by analyzing their legal basis, an evaluation has been performed from the perspective of planned development of urban structures of the subject coastal settlements. The results of the research are presented in three subsections: analysis and comparison of the transformation of urban structures of the coastal settlements of Boka Kotorska; identification of the features and elements that affect planned development of the settlements; evaluation of planning criteria and models through previously systematized physical planning documentation, and proposal of new planning criteria and models for improving coastal settlements which can be used in practice, i.e., through the preparation of physical planning documentation, studies and expertise.

### 3.1. Analysis and Comparison of the Transformation of Urban Structures of the Subject Coastal Settlements of Boka Kotorska during the Observed Period

The analysis and comparison of the identity attributes of the subject coastal settlements of Montenegro was performed considering physical planning documents, state maps and statistical data prepared during the observed period. A comparison and graphic presentation of the area of built-up land (BL) throughout the two observed periods, considering the share of BL area in the settlement area, distribution of BL according to altitude (up to 100 m above sea level and over 101 m above sea level) as well as occupation of BL in relation to the coastline (distance band up to 100 m, and over 101 m), is shown in Figure 4 with the examples of three coastal settlements in the municipalities of Kotor, Tivat and Herceg Novi. Red areas in Figure 4 represent built-up land in the first observed period, gray areas in the second observed period, lines (red) represent the distance of 100 m from the coastline, brown lines the altitude of 100 m above sea level and black lines the road traffic.

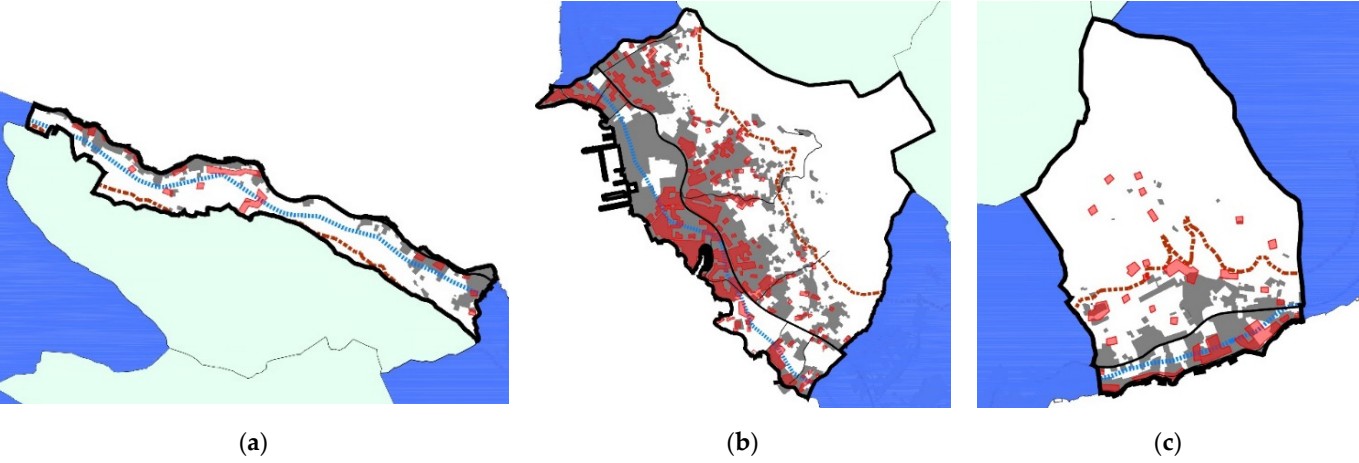

|   (a)   |   (b)   |   (c)   |

**Figure 4.** Depiction of the built-up land of the coastal settlements during the two observed periods: (**a**) coastal settlement Donji Stoliv (Kotor municipality); (**b**) coastal settlement Tivat (Tivat municipality); (**c**) coastal settlement Baošići (Herceg Novi municipality).

In order to compare distinctive elements of coastal settlements, a cartogram was used, which shows the built-up land in two different time periods (1945–1992, and the situation in 2006). For the first observed period, the maps that were available during the preparation of the paper were used and are noted in Table 2. A comparison of changes in built-up land of the coastal settlements of Boka over the two observed periods was made on the basis of the cartogram of the existing state in 1970, i.e., pursuant to the General Urban Plan of Boka Kotorska (Kotor municipality [80], Tivat municipality [42], Herceg Novi municipality [52]). The spatial planning document in question consists of the existing and planned state. The

geodetic cadastral base created in 1970 for the purposes of the General Urban Plan is also part of the existing state.

As for the other observed period, cadastral maps of the Real Estate Administration of Montenegro digitalized between 2004 and the proclamation of independence (2006) were used. These results were obtained by overlaying the cadastral maps and the cartogram of the existing state in 1970 in AutoCAD (AutoCAD Map). Overlapping was carried out on the basis of built-up cadastral plots through two different periods, that is, for each cadastral plot that was built, regardless of whether it is in accordance with the PPD or not. The share of the built-up land of coastal settlements in relation to coastal municipalities and the observed period is shown in Table 3.

**Table 3.** The share of BL (built-up land) in the coastal settlements of Boka Kotorska during the observed period.

| Municipality | Area of the Coastal Settlements (ha) | Area of BL in the First Observed Period (ha) | Area of BL in the Second Observed Period (ha) | The Percentage of the Increase (%) |
|---|---|---|---|---|
| Kotor | 10,775 | 205 | 448 | 2.25% |
| Tivat | 3605 | 123 | 518 | 10.95% |
| Herceg Novi | 6399 | 252 | 641 | 6.08% |
| Σ | **20,779** | **580** | **1607** | **4.94%** |

The features of the coastal settlements (built-up land) have been analyzed from two perspectives: with reference to the coastline (in order to prove the assumption of a stronger impact of built-up land in the zone of 100 m distance from the coastline) and in relation to elevations (to prove the assumption that in the midst of excessive construction along the coast, the built-up land expands even to elevations above 100 m above sea level; to prove another assumption that old urban structures at higher elevations tend to collapse or become abandoned). For the above elements analyzed as one of the features of the change in the urban structure of Boka from 1945 to 2006, the abbreviations shown in Table 4 are used in this paper.

**Table 4.** Meaning of the abbreviations used in the subject research and analysis of the change in the area of the built-up land over two observed periods is presented in more detail in relation to the coastline and the altitude (100 m above sea level) for Kotor municipality.

| Abbreviation | Meaning |
|---|---|
| A | Increase /decrease in the area of built-up land of the coastal municipality, in relation to the area of the municipality, observed over two periods, expressed in percentage |
| B | Increase/decrease in the area of built-up land of the coastal settlement, in relation to the area of the municipality, observed over two periods, expressed in percentage |
| $C_1$ | Increase/decrease in the area of built-up land within the coastal zone of the municipality (up to 100 m distance from the coastline to the mainland), in relation to the area of the municipality, observed over two periods, expressed in percentage |
| $D_1$ | Increase/decrease in the area of built-up land within the coastal zone of the settlement (up to 100 m distance from the coastline to the mainland), in relation to the area of the settlement, observed over two periods, expressed in percentage |
| $C_2$ | Increase/decrease in the area of built-up land outside the coastal zone of the municipality, in relation to the area of the municipality, observed over two periods, expressed in percentage |
| $D_2$ | Increase/decrease in the area of built-up land outside the coastal zone of the settlement, in relation to the area of the municipality, observed over two periods, expressed in percentage |
| $E_1$ | Increase/decrease in the area of built-up land in the municipality up to 100 m above sea level, in relation to the area of the municipality, observed over two periods, expressed in percentage |
| $F_1$ | Increase/decrease in the area of built-up land in the settlement up to 100 m above sea level, in relation to the area of the settlement, observed over two periods, expressed in percentage |
| $E_2$ | Increase/decrease in the area of built-up land in the municipality of 100 m above sea level, in relation to the area of the municipality, observed over two periods, expressed in percentage |
| $F_2$ | Increase/decrease in the area of built-up land in the settlement of 100 m above sea level, in relation to the area of the settlement, observed over two periods, expressed in percentage |

**Table 4.** *Cont.*

| Coastal settlement | A | B | C₁ | D₁ | C₂ | D₂ | E₁ | F₁ | E₂ | F₂ |
|---|---|---|---|---|---|---|---|---|---|---|
| **Kotor municipality** | | | | | | | | | | |
| Kotor | | +37.06 | | +29 | | +8.05 | | −4.94 | | 0 |
| Bigova | | +13.04 | | +9.28 | | +3.75 | | +13.03 | | 0 |
| Glavatići | | −0.27 | | +0.39 | | −0.65 | | +0.12 | | −0.39 |
| Gornji Orahovac | | −0.08 | | 0 | | −0.08 | | 0 | | −0.07 |
| Dobrota | | +9.87 | | +4.50 | | +5.36 | | +9.82 | | +0.04 |
| Donji Morinj | | 0 | | +0.62 | | -0.62 | | +0.82 | | −0.82 |
| Donji Orahovac | | +23.93 | | +10.97 | | +12.95 | | +23.93 | | 0 |
| Donji Stoliv | | +17.08 | | +17.08 | | 0 | | +15.40 | | +1.68 |
| Dražin Vrt | | +2.13 | | +5.60 | | −3.46 | | +2.13 | | 0 |
| Zagora | | −0.50 | | 0 | | −0.50 | | 0 | | −0.50 |
| Kostanjica | +2.25 | +0.96 | +4.63 | +2.36 | +1.91 | −0.41 | +4.45 | +1.25 | +0.05 | −0.30 |
| Krimovica | | +1.91 | | +0.06 | | +2.28 | | +0.21 | | +2.13 |
| Lipci | | +4.97 | | +4.61 | | +0.35 | | +4.97 | | 0 |
| Lješevići | | +1.38 | | 0 | | +1.37 | | +1.30 | | +0.07 |
| Muo | | +6.46 | | +5.69 | | +0.70 | | +6.45 | | 0 |
| Perast | | +0.27 | | +0.25 | | +0.01 | | +0.25 | | +0.01 |
| Prčanj | | +4.91 | | +3.71 | | +1.19 | | +4.91 | | 0 |
| Risan | | +2.90 | | +1.04 | | +1.84 | | +3.59 | | −0.69 |
| Strp | | +0.52 | | +0.78 | | −0.26 | | +0.52 | | 0 |
| Višnjevo | | −0.77 | | +0.01 | | −0.78 | | +0.01 | | −0.78 |
| Škaljari | | +10.55 | | +1.37 | | +9.19 | | +9.83 | | +0.73 |
| **Tivat municipality** | | | | | | | | | | |
| Coastal settlement | A | B | C₁ | D₁ | C₂ | D₂ | E₁ | F₁ | E₂ | F₂ |
| Tivat | | +27.87 | | +6.30 | | +23.58 | | +29.18 | | +0.71 |
| Bogišići | | −0.43 | | −0.98 | | +0.56 | | −0.75 | | +0.33 |
| Gošići | | +3.05 | | +0.64 | | +2.40 | | +1.99 | | +1.05 |
| Donja Lastva | | +9.99 | | +3.61 | | +6.37 | | +10.35 | | −0.34 |
| Đuraševići | | +5.41 | | −0.09 | | +5.50 | | +5.37 | | +0.03 |
| Krašići | +10.95 | +10.00 | +1.86 | +4.52 | +7.22 | +5.47 | +8.92 | +9.57 | +0.16 | +0.42 |
| Lepetani | | +1.77 | | +2.50 | | −0.72 | | +1.81 | | −0.04 |
| Milovići | | +2.83 | | +0.39 | | +2.42 | | +3.82 | | −1.00 |
| Mrčevac | | +29.32 | | +1.74 | | +27.53 | | +29.15 | | +0.12 |
| Radovići | | −0.87 | | +0.03 | | −0.90 | | −1.26 | | +0.39 |
| **Herceg Novi municipality** | | | | | | | | | | |
| Coastal settlement | A | B | C₁ | D₁ | C₂ | D₂ | E₁ | F₁ | E₂ | F₂ |
| Herceg Novi | | +36.73 | | −0.42 | | +37.16 | | +32.70 | | +4.00 |
| Baošići | | +9.85 | | +2.47 | | +7.37 | | +10.31 | | −0.47 |
| Bijela | | +12.22 | | +2.35 | | +9.87 | | +4.11 | | +0.76 |
| Đenovići | | +14.85 | | +5.36 | | −2.84 | | +15.70 | | −0.85 |
| Đurići | | +3.98 | | +2.26 | | +1.71 | | +5.06 | | −1.08 |
| Igalo | | +42.40 | | +5.77 | | +36.61 | | +42.39 | | 0 |
| Jošice | +6.08 | +5.25 | +2.98 | +4.56 | +10.80 | +3.89 | +13.75 | +8.83 | +0.33 | −0.37 |
| Kumbor | | +11.28 | | +5.82 | | +6.72 | | +11.68 | | −0.40 |
| Luštica | | +0.01 | | +0.09 | | −0.08 | | +0.18 | | −0.16 |
| Meljine | | +15.56 | | +2.11 | | +13.44 | | +15.56 | | 0 |
| Provodina | | +4.52 | | +1.07 | | +3.44 | | +3.77 | | +0.73 |
| Sasovići | | +3.59 | | +0.84 | | +2.75 | | +1.44 | | +2.15 |
| Zelenika | | +27.13 | | +6.54 | | +20.59 | | +27.13 | | 0 |

The results of the analysis of the change in the area and position of the built-up land over two observed periods in vertical and horizontal dimensions, depending on the distance from the coastline (100 m towards the mainland) and the altitude (100 m above sea level) for all coastal settlements of Boka, for each settlement separately, are shown in Table 4. The results of the conducted research show that the reduction of built-up land is a consequence of the demolition of buildings in the observed period. The demolition of buildings took place in the area of the settlement above 100 m above sea level, as well as outside the coastal zone. These data confirm the greater pressure of the expansion of built-up land in the settlement area within 100 m of the coastline, and the "extinction" of part of the settlement in the hinterland of the municipality.

The presented results confirm one of the initial research hypotheses, i.e., they prove the assumption that urban transformations occurring in coastal settlements of Boka Kotorska over the observed period took place mostly during the second observed period (after 1992), as graphically presented in Figure 5.

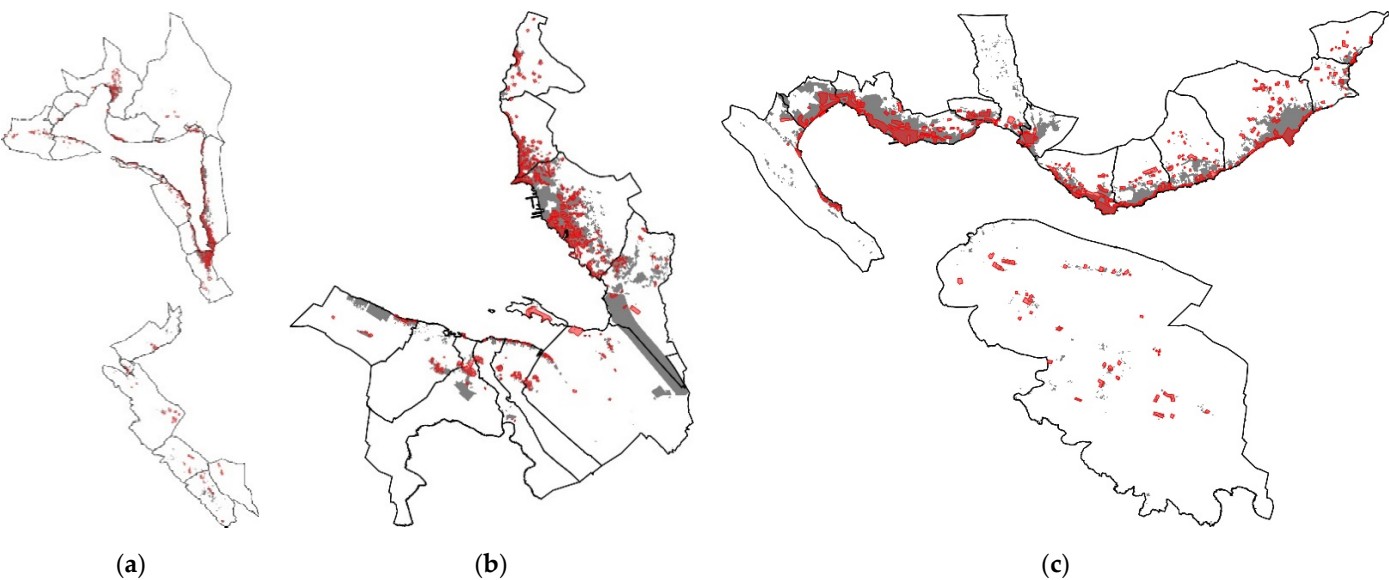

(**a**)                                              (**b**)                                              (**c**)

**Figure 5.** Urban pattern of the development of Boka in the 1970s: (**a**) Kotor municipality; (**b**) Tivat municipality; (**c**) Herceg Novi municipality. The area marked red represents the built-up land in the first observed period, and the area marked gray represents the built-up land from 2006.

### 3.2. Distinctive Elements and Features of Spatial Development of the Coastal Municipalities of Kotor, Tivat and Herceg Novi

Upon systematizing planning material for 42 coastal settlements in Boka Kotorska, their features have been identified and analyzed as a condition for their further planning development, through grouping of natural features, features of cultivated landscape and distinctive elements of the built-up (cultural–historical) environment.

### 3.2.1. Natural Features of the Municipalities of Kotor, Tivat and Herceg Novi

The natural features of the subject area were analyzed in the physical planning documentation from the aspect of geomechanical and pedological features, relief and other terrain characteristics. Geographical, geological, hydrographic, hydrogeological, pedological and climatic characteristics are natural factors that influence the formation and development of settlements [85]. In addition, natural conditions of an area also include "seismological characteristics and phenomena, geomorphological characteristics of phenomena and conditions, and climatic characteristics, which determine the state and values of an area" [83].

Natural features that, among others, influence planned development of coastal areas of Boka Kotorska the most are relief and other characteristics of the terrain. Terrain configuration and pedological characteristics of Adriatic coastal settlements have been analyzed using the coast indentation index, terrain slopes, types of beaches, their shape and area as well as the existence of islands along coastal settlements. Coast indentation may be observed horizontally or vertically. The shape of the coastline can be mathematically expressed using coast indentation index (If), which is the ratio of the length of the coastline L and the length of the straight line that joins the first and the last point of the coastline of the settlement D (Euclidean length) [86], as graphically shown in the example of coastal settlement of Prčanj (Figure 6):

$$If = L : D, \tag{1}$$

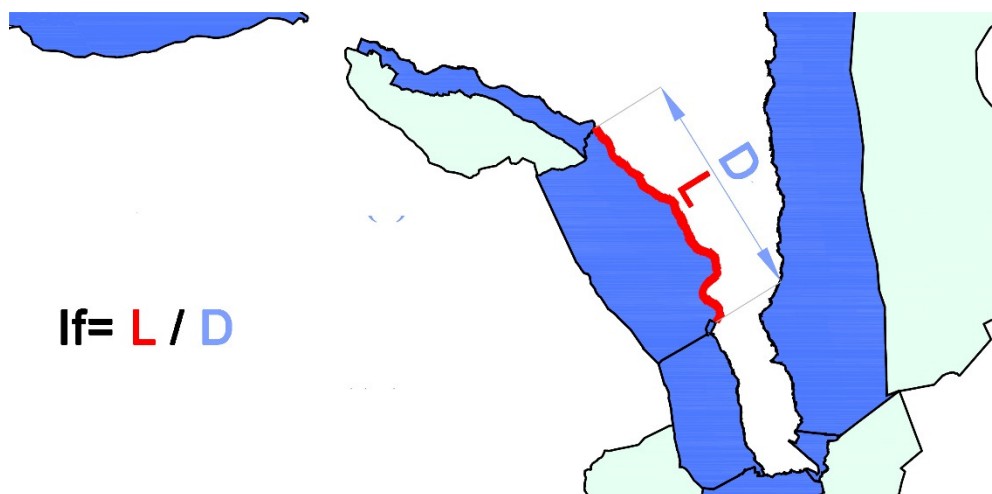

**Figure 6.** Example of how to calculate indentation index for the coastal settlement Prčanj (Kotor).

Horizontal indentation index of the coastline depends on the terrain topography, i.e., on the occurrence of coves (inlets), as well as capes, reefs and rocks. Horizontal indentation index [87] is shown in Table 5 for each coastal municipality and coastal settlement of Boka, with the aim of defining the criteria for their planned development based on the shape of the coastline.

**Table 5.** Coast indentation index of coastal municipalities and settlements of Boka Kotorska.

| Kotor Municipality | | | | | | | | | |
|---|---|---|---|---|---|---|---|---|---|
| Kotor | 1.80 | Bigovo | 1.50 | Glavatići | 2.17 | G. Orahovac | 1.05 | Dobrota | 1.89 |
| Donji Morinj | 1.50 | D. Orahovac | 1.66 | Donji Stoliv | 1.36 | Dražin Vrt | 1.19 | Zagora | 1.55 |
| Kostanjica | 1.28 | Krimovica | 2.25 | Lipci | 1.24 | Lješevići | 1.34 | Muo | 1.55 |
| Perast | 1.42 | Prčanj | 1.92 | Risan | 3.36 | Strp | 1.20 | Višnjevo | 2.44 |
| Škaljari | 1.39 | | | | | | | | |
| **Tivat municipality** | | | | | | | | | |
| Tivat | 2.92 | Bogišići | 4.90 | Gošići | 2.62 | D. Lastva | 1.41 | Đuraševići | 1.20 |
| Krašići | 1.36 | Lepetani | 1.47 | Milovići | 2.07 | Mrčevac | 1.64 | Radovići | 2.53 |
| **Herceg Novi municipality** | | | | | | | | | |
| Herceg Novi | 1.65 | Baošići | 1.61 | Bijela | 1.45 | Đenovići | 1.70 | Đurići | 1.21 |
| Igalo | 1.32 | Jošice | 1.60 | Kumbor | 1.60 | Luštica | 4.39 | Meljine | 1.12 |
| Provodina | 1.44 | Sasovići | 1.16 | Zelenika | 1.74 | | | | |

Vertical indentation (unevenness) in the subject area is most pronounced in the Bay of Kotor and Risan (coastal settlements of Dobrota–Orahovac), with a horizontal distance of 1.5 km an altitude difference of 1050 m [88]. Vertical and horizontal indentation affects the shape, size and use of parts of the construction area, and are closely related, i.e., the topography of the beach terrain is reflected in the topography of the sea terrain. Thus, along the rocky beaches—the sea is deep, and along the low beaches—the sea is shallow. The natural features of the beach can be described using the following terms [86], as shown in the Table 6.

**Table 6.** Quotients of the shore and their mathematical formula.

| Quotient | Ratio (Formula) |
|---|---|
| Maritimity | Length of developed seashore (km)/Area of municipality (km$^2$) |
| Indentation | Length of developed shore (km)/Linear length of the shore (km) |
| Swimming Beach | Length of the shore used for swimming (km)/Length of developed shore (km) |
| Littotality | Area of the shore belt (500 m) (km$^2$)/Area of municipality (m$^2$) |

### 3.2.2. Features of the Cultivated Landscape of Boka Kotorska

Cultivated landscape also refers to natural and environmental occurrences that include mostly olive groves, vineyards and sporadically orchards, gardens and other arable crops [22]. Authentic landscape is one of the distinctive elements and characteristics of the origin and development of coastal settlements of Montenegro. By analyzing their landscape, three categories have been identified based on valuation of types of landscape character in the coastal zone [3], namely: "natural and semi-natural landscapes" (marine water area, forests, brushwood and forest land and water surfaces), "cultural landscapes" (ambience units and architectural heritage) and "special agricultural landscapes" (flattened fields of alluvial and alluvial–colluvial soil; terraces and plateaus on flysch and karst terrain; special agricultural areas, important for preservation of cultural heritage and landscape characteristics, developed as a result of application of traditional procedures in cultivation and maintenance of agricultural soil, etc.).

The economic and landscape value of olive groves in the area of coastal settlements is part of the autochthonous plant inventory and traditional mosaic of the Montenegrin coast. Olive groves are defined as "indisputable objects of nature protection of exceptional value in the entire Mediterranean". Moreover, traditional terraces with olive groves have been recognized as one of the 26 types of landscape character, that is, the bearers of the identity and recognizability of the settlement [29]. Spacious olive groves with traditional terraces that participate in the valorization of the coastal landscape are located in the coastal settlements of Glavatići, Donji Morinj, Kostanjica (Kotor municipality), Luštica, Meljine and Sasovići (Herceg Novi municipality). These interpolated vineyards situated on terraced slopes have been recognized as an opportunity and a way to improve and enrich even the most deserted (most degraded) part of the landscape [29]. In addition, extensive plantations of orchards and vineyards in the hinterland of Herceg Novi have been noted.

The devastation of the environment (especially natural) occurred during the Middle Ages as a result of the sudden devastation of forests in coastal mountain areas. As a consequence of deforestation, deposits of silt and gravel landfills disabled ports, especially those at the estuary where a confluence of rivers streams into the sea, by changing the shape of the coastline, and thus changing natural conditions. This cycle of natural changes is still noticeable today in the coastal settlements of Donji Morinj, Krimovica, Lipci, Višnjevo (Kotor municipality) and Radovići (Tivat municipality). In addition to the abovementioned types of landscape character, forest slopes on flysch and diluvium, bare hilly terrains on limestones, coastal ridges, rocky shores, and coastal and flooded alluvial plains and wetlands (coastal settlements of Lješevići, Mrčevac, Đuraše) were identified. The above features should be one of the main conditions for defining planning criteria and models for the development of coastal settlements in Boka, regardless of whether they are devastated areas or other forms of natural environmental features.

### 3.2.3. Distinctive Elements of the Built-Up (and Cultural–Historical) Environment

The built-up land of the coastal settlements of Montenegro has been observed through the prism of analyzing the network of settlements, population distribution, transport connectivity, economic characteristics and valuation of cultural assets. The analysis of the network of settlements from the first observed period, based on the Regional Spatial Plan of Montenegrin Littoral from 1966 [19], has been conducted according to the size, i.e., the number of inhabitants (under 600; between 600 and 1500; between 1500 and 3000; between 3000 and 30,000; over 30,000). On the other hand, the classification for the second observed period [28] includes 10 categories: uninhabited settlements; settlements with 1–25 inhabitants; with 26–50; with 51–100; with 101–500; with 1000–2000; with 2000–4000; with 4000–10,000; with 10,000–13,500 inhabitants (Figure 7).

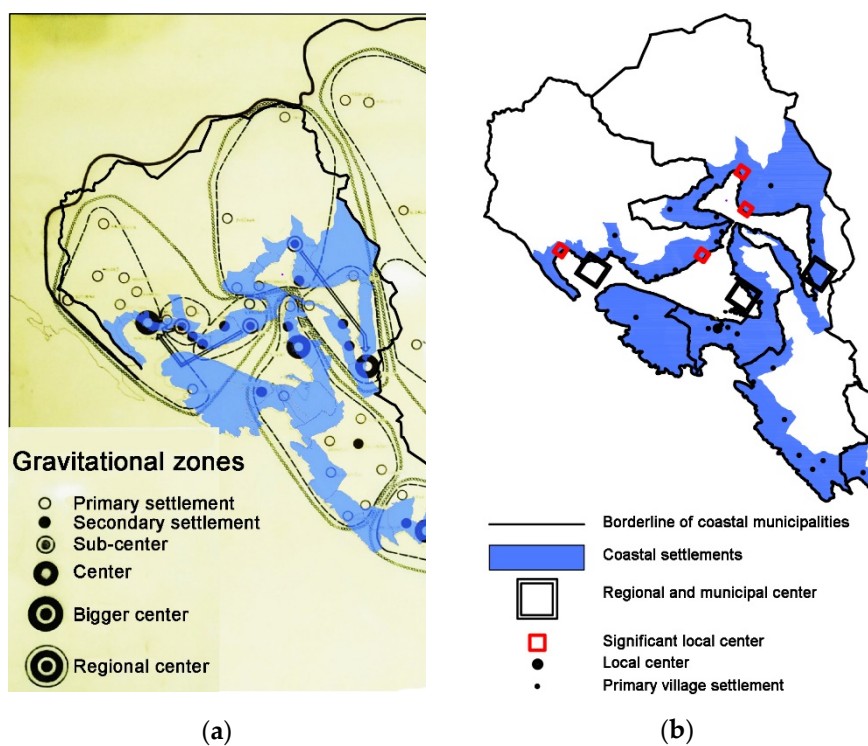

**Figure 7.** Classification of settlements in the Coastal Zone of Montenegro according to their size, over two observed periods: (**a**) in 1966; (**b**) in 2018.

The feature that contributes to the distinctiveness of a coastal settlement is a demographic feature of a settlement, i.e., a shift in the number of inhabitants along the coast. In order for the settlement to be sustainable, the minimum number of inhabitants defined on the basis of systematization of inhabitants [21] should be from 50 to 100. Out of the 44 settlements in question, this condition is not met by 5 coastal settlements of Kotor (Gornji Orahovac, Donji Orahovac, Zagora, Lipci, Strp), according to the 2003 census [74]. The demographic analysis of the coastal settlements of Boka has been carried out for both observed periods (1948–1992 and 1992–2006) by examining the littoralization of the population ("quotient of the population of the coastal zone and the number of inhabitants" [89]). In Figure 8, the colors indicate the changes in the demographic characteristics of the settlements according to municipalities as follows: black indicates the coastal settlement that had the largest increase in population in the observed period compared to the municipality to which it belongs; green indicates the coastal settlement that had a decrease in population compared to the municipality, over two different time periods; red indicates the coastal settlement for which in both observed periods the increase in population compared to the coastal municipality is noted; blue indicates the settlement that had a stagnation or possible slight increase/decrease in population.

Based on the results analyzed over two periods, and in relation to the number of inhabitants of the settlement in relation to the number of inhabitants of the municipality (littoralization) and based on the tabular representation of the increase/decrease in the built-up area of the land of the coastal settlements of Montenegro, in Table 7, the relationship between two distinguishing features of the settlement (built-up area and number of inhabitants) is shown. Under the category of rest in the expansion of built-up (or built-up land at rest), this research treated every coastal settlement of Montenegro whose percentage of increase/decrease in the area of built-up land in the observed period is up to and below 1%. Other settlements are settlements with an increasing area of built-up land.

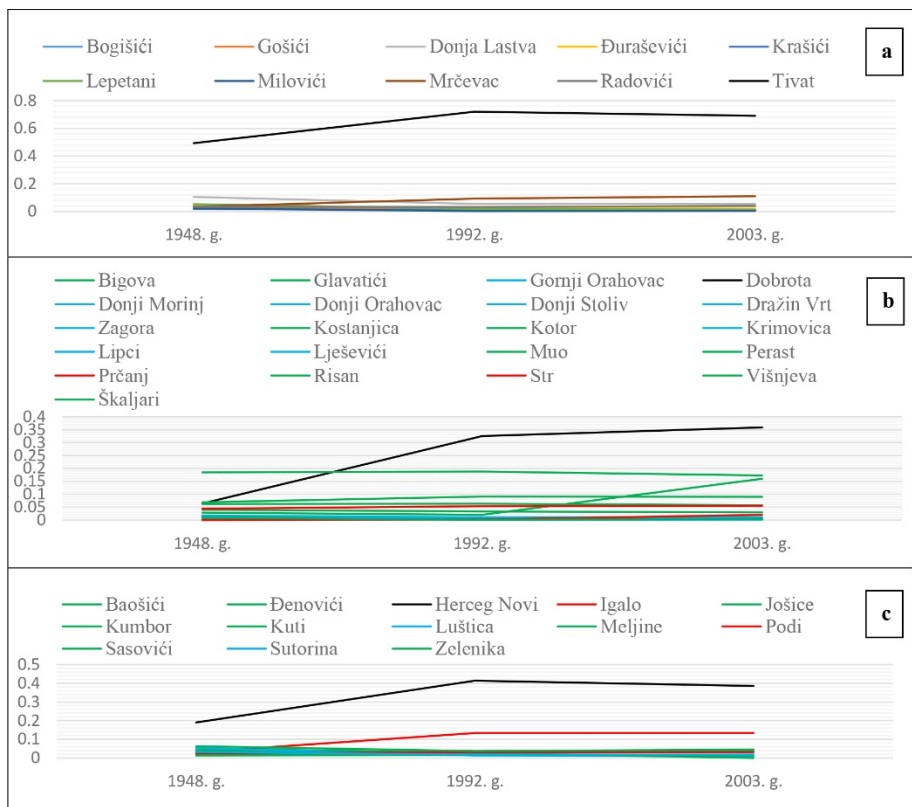

**Figure 8.** Analysis of the littoralization of the coastal settlements of Kotor municipality (**a**), Tivat (**b**) and Herceg Novi (**c**) in both observed periods.

**Table 7.** Tabular presentation of the relationship between the built-up area and the number of inhabitants during the two observed periods.

| Number of Inhabitants in the Settlement in Relation to the Municipality | Built-Up Area in Increase | Rest in the Spread of the Built-Up Area |
|---|---|---|
| No. of settlements with the largest number of inhabitants | 3 | / |
| No. of settlements with an increase in the number of inhabitants | 5 | / |
| No. of settlements with declining population | 17 | 4 |
| No. of settlements with rest no. inhabitants | 12 | 3 |

Based on the results obtained through graphs and tables, it can be concluded that the increase in the number of inhabitants within the coastal settlement in relation to the municipality does not mean an increase in the built-up land area of the settlement. These data speak of the increased influence of the tourist aspect on the development of the coastal zone and the formation of the so-called tourist settlements.

In the first observed period, according to spatial organization of the coastal region of Montenegro into "zones, settlements, and centers" [78], and because the centrality of the coastal settlement of Boka Kotorska is defined according to the classification of central functions, size of influential areas and size of centers [21], the municipality of Tivat is defined as a center of industry and air transport, the municipality of Kotor as a business, scientific and cultural center, the municipality of Herceg Novi as a tourist center specializing in health care tourism with cultural functions, and the coastal settlement of Luštice as a new tourist center of Herceg Novi. The natural features of the area, such as the topography of the terrain, greatly influenced the classification of the settlements according to the centrality. Thus, the coastal settlement of Kotor had less intense gravitational force compared to other coastal settlements of Montenegro, given that its hinterland is extremely vertical and

does not allow the possible construction and expansion of the construction area toward the mainland.

The lack of space for shaping public functions along the coastline of the municipality of Herceg Novi was criticized in the 1980s in the physical planning documentation [89] addressing the construction of individual (weekend) housing constructions along the coastal belt and the failure to integrate major architectural units in the scenery of the Bay of Boka Kotorska. In the 1980s, the eastern part of the coastal settlement of Igalo was notorious for the extremely illegal construction of the area, i.e., connecting the construction areas of the coastal settlements of Igalo and Herceg Novi was noticed. In the coastal municipality of Tivat, in the period between 1960 and 1980, most of the facilities in the coastal settlements of Donja Lastva, Tivat and Mrčevac were built according to physical planning documentation [86].

Distinctive elements of the built-up area of coastal settlements of Boka Kotorska in the period 1945–2006 have been analyzed in light of four aspects of transport connectivity: road, rail, air and maritime transport.

Road transport of Boka Kotorska did not change significantly compared to the first observed period, i.e., it had been defined to a certain extent even before 1945 (construction of the road Kamenari-Risan-Kotor 1965–1966; Debeli Brijeg-Zelenika 1964–1966; Zelenika-Kamenari 1964–1965, Lepetani-Tivat-Radanovići 1964–1965) [90].

Until 1963, when, upon the decision of the Commission of the Federal Executive Council 1958–1959, the Adriatic highway was built, the economy of the coastal settlements of Boka had been poor (agricultural land had been limited, the industry had been underdeveloped), and consequently, the economic development of the coastal settlements had stagnated. With the construction of the highway, tourism became a strong prerequisite for their urban development. Railway traffic in the Boka Kotorska dates back to the beginning of the 20th century, in the area of the coastal settlement of Zelenika (Herceg Novi municipality), and in the second observed period, this type of traffic was no longer active in the subject area. The main airport of the Coastal Region of Montenegro is located in the coastal settlements of Mrčevac and Đuraševići (coastal municipality of Tivat). The higher-level physical planning documentation adopted after World War II [15,78,79] projected keeping the airport. According to the Regional Spatial Plan of Montenegrin Littoral from 1966 [19], all municipalities of the Bay of Boka Kotorska were connected by maritime lines. Further connection of the passenger maritime system with the rest of the coastal settlements of Montenegro was established via Budva, Petrovac, Bar and Ulcinj, and freight transport via coastal settlements of Budva, Bar and Ulcinj. The construction of facilities of maritime traffic were planned in coastal settlements in the Kotor municipality, i.e., in the coastal settlement of Bigova, where the construction of a sports port and a pier were planned.

One of the dominant elements of the physical structures of coastal settlements are hiking trails (urban hiking trails, and promenades or tourist esplanades), designed for up to 4–5 km, with high-quality tourist attractions (diverse landscape, recreation areas, sightseeing platforms, easy access, round trips, roofed pathways and shaded rest stops) [91]. The need for a better solution regarding the transport system (especially road transport) arose in the second observed period due to the increase in economic flows (tourism pressure) in the coastal region of Montenegro.

Elements of distinctiveness of the built-up area of the coastal settlements of Boka Kotorska also include economic characteristics observed through the following aspects: economy (tourism, agriculture, marine fisheries and mariculture, shipyards and ship repair, shipping) and industry (crude oil and gas extraction, construction) [29]. In the period up to the second half of the 20th century, the economic criterion for land valuation was closely related to agriculture as a natural resource and the basis of lifestyle choices. Both before and after the observed period (1945), shipbuilding and industrial complexes (in the coastal settlements of Tivat, Kotor and Bijela) were dominant in coastal settlements. After 1945, industrialization caused the increase in the number of employed people, migration within

the state, the change in the economic and urban structure of the area as well as the change of the standard of living.

In the first observed period, 15% of all employed people in Herceg Novi municipality worked in industry—mostly in the shipyard for ship repair in the coastal settlement of Bijela, while in the municipality of Tivat, the largest number of employed people worked in industry (43.6%) [86]. After 1945, other natural potentials were identified that could have affected the economic sector of the settlement or municipality. For example, the sandy part of the beach in the length of 1700 m in the coastal settlement of Igalo (Herceg Novi municipality) was recognized due to its natural features for its tourist–health (economic) potential and the use of sand (mud) for health purposes. Both in the first (1966) and after the second period (2018), tourism accounted for over 50% of total revenues at the level of the coastal region of Montenegro, and thus was the main economic feature of the development and improvement of coastal settlements.

Valuation of cultural properties aimed at recognizing criteria for development and improvement of the area they belong to may be performed considering the characteristics of individual objects and areas [92]. Pursuant to the Law on the Protection of Cultural Property of Montenegro [93], the types of immovable cultural properties are: a cultural and historical structure, a cultural and historical whole (an old town, an urban or rural settlement, a homogenous cluster of buildings) and a locality or area (an archeological find, a cultural and historical area, and a cultural landscape—"a space whose characteristic appearance is the result of an action and interaction between natural and antropogenous factors in a longer span of time"). Two zones have been identified in the subject area in terms of landscape and ambient characteristics: the first is the territory of Boka Kotorska and the second is the territory from Tivat to Jaz [92]. A special area is the Protected Environment of the Natural and Cultural–Historical Area of Kotor, acknowledged by UNESCO, which has an exceptional universal value as a World Heritage Site [70].

The analysis of listed cultural monuments from 1966 (total 104) [78] and the protected cultural properties from 2018 (total 307) (Directorate for the Protection of Cultural Heritage) shows that the number of cultural properties increased significantly (Figure 9), i.e., the list of the protected cultural properties was notably expanded.

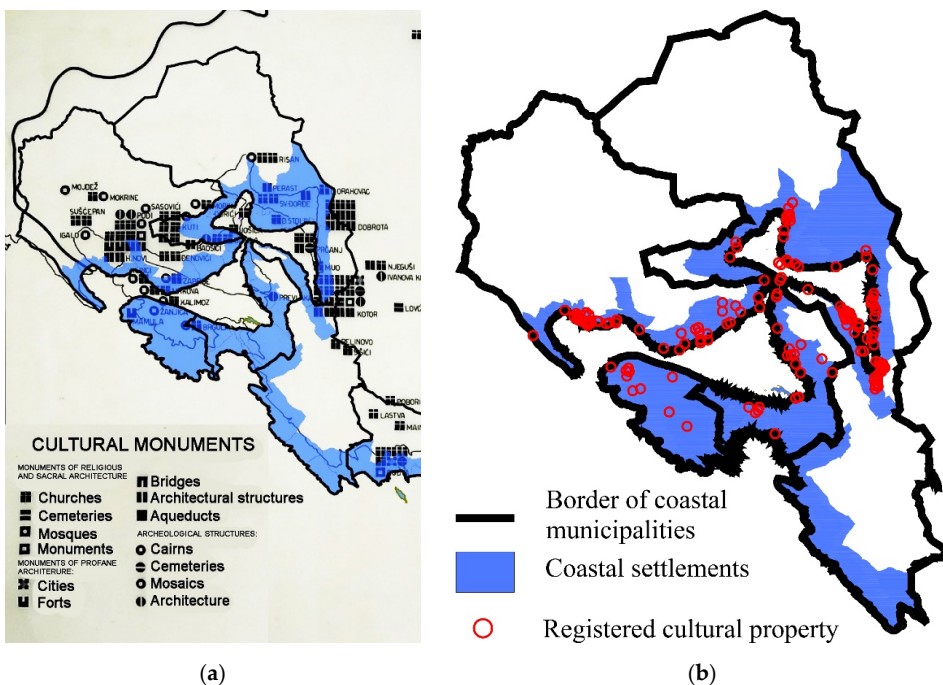

(a)                                           (b)

**Figure 9.** Characteristics of the cultural–historical environment of the Coastal Region of Montenegro over two observed periods: (**a**) Cultural monuments form 1966. (**b**) Cultural properties from 2018.

## 4. Discussion

On the basis of the obtained results, in the continuation of this research, those planning criteria for the development and improvement of coastal settlements will be determined when they are based on the grouping of the mentioned identity characteristics, that is, those characteristics that are recognized as elements of the development of coastal settlements. Based on the recognized criteria for the improvement and development of coastal settlements, general and special models for the development of the subject area will be presented, recognized from the available, analyzed and evaluated PPDs. Through their analysis, their additions will be described and proposed.

### 4.1. Identified Planning Criteria for the Improvement of Coastal Settlements of Boka Kotorska

Primary criteria for the spatial development of coastal settlements of Boka were identified and analyzed in this study. The criteria refer to: criteria for natural values, cultural–historical, legislative, spatial–urban and economic criteria. In addition, it includes the following planning models: model for development and improvement of coastal settlements (preservation of natural values of the identity of coastal settlements through preservation of "OF" and "FROM" views), infrastructural model of growth of urban development of coastal settlements, economic model, spatial–urban model of sustainable development and model of preservation and protection of historical parts of settlements.

Upon the systematization of the physical planning documentation (25 documents), the criteria for the improvement and development of the coastal settlement or part of the coastal settlement, defined on the basis of their distinctive features, were identified. The following acronyms are used to represent the identified criteria: criterion for natural values—CNV, cultural–historical criterion—CHC, legislative criterion—LC, spatial-urban criterion—SUC and economic criterion—EC. Moreover, a proposal for the planned development of a coastal settlement or part of a settlement and municipality is given in Table 8.

**Table 8.** Identified criteria for planned development of coastal settlements in Boka Kotorska based on their distinctive elements and features through physical planning documentation.

| | Criteria | Distinctive Elements and Features of Coastal Settlements in Boka | | | Proposal for the Planned Development | Settlement/Part of a Settlement (Municipality) |
|---|---|---|---|---|---|---|
| **1** | CNV; SUC | Nonbuilt part of a construction area | | Unfit for construction due to geological and morphological characteristics | Agricultural development | Western part of Igalo (Herceg Novi) |
| **2** | CNV; SUC; EC | Terrain topography (slope) | Pine forest complex | Orientation (south-east) | Hotel accommodation | Pržna, Bigova (Tivat) |
| | | Nonbuilt part of a construction area | | Visual connection with the sea and the mainland | | |
| | | Terrain topography (flat terrain) | Olive groves | Beach accessibility | | |
| **3** | CNV; SUC | Terrain configuration | | Insolation | Construction of a tourist complex | Markov Rt, Prčanj (Kotor) |
| **4** | CNV | Connecting the area with the mountain and the sea | | Necessary area for expanding the capacities | Health care-tourist center | Igalo (Herceg Novi) |
| | | Panoramic view of the Bay of Boka | | Climatic conditions | | |
| **5** | CHC; SUC | Structure of a construction area: scattered | Proximity of the shore | Historical properties | Polycentric development of settlements | Luštica (Herceg Novi) |
| | | Partially nonbuilt part of a construction area | | Distinctive elements of cultivated landscape (olive groves) | | |
| **6** | CHC; SUC | Distinctive elements of cultivated landscape | | Preservation of panoramas | Road construction | Đurići, Jošice, Lepetani (Herceg Novi-Tivat) |
| | | Distinctive elements of a cultural–historical area | | | | |
| **7** | CNV; SUC | Terrain topography | | Insolation | Construction of a specific residential–tourist point | Morinj (Kotor) |
| | | Visual perception | | | | |

Nonbuilt area of the western part of the coastal settlement of Igalo was considered to be unfit for construction during the 1980s due to its geological and morphological characteristics. Therefore, it is proposed that the subject area is intended for agriculture (number 1 in Table 8).

The analysis of natural features of coastal settlements influenced the determination of the type of a tourist accommodation and its positioning along the coastline in the coastal settlement of Bigova, Kotor municipality (number 2 in Table 8). Terrain topography in physical planning documentation of the costal settlement of Bigova [50] caused the positioning of hotel accommodation at higher elevations due to: greater and stronger visual connection with the environment; (vertical) distancing from the coastline and thus liberating the accessible part of the beach from construction pressure; activating the flora (pine forest complex) in order to create a more intimate area; and orientation (southeast) (marked by (a) in Figure 10). This document anticipated the construction of campsites and recreational facilities along the coastline, in the area where the terrain is slightly sloping and where olive groves are present, which would adapt to the indented natural environment (marked by (b) in Figure 10). The physical planning documentation projected a health care, hospitality and administration facility in the entrance zone of the planned complex (marked by (c) in Figure 10).

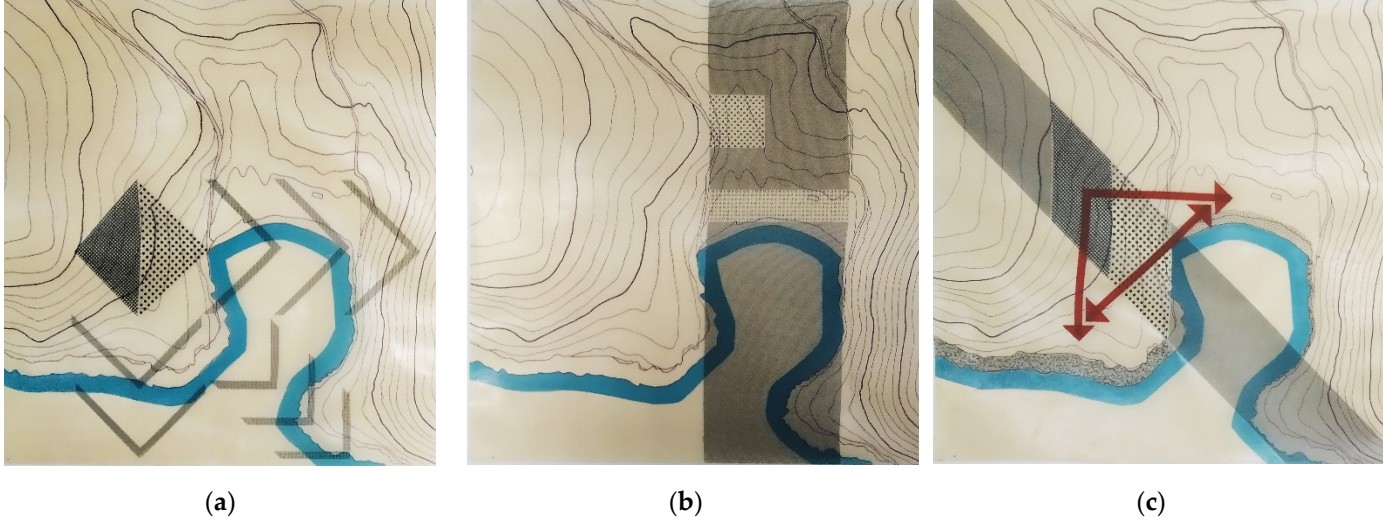

(**a**)　　　　　　　　　　　　　　(**b**)　　　　　　　　　　　　　　(**c**)

**Figure 10.** Physical planning document of Pržno from 1969: (**a**) analysis of the relationship between the elements and the views in space; (**b**) positioning the camp and the recreational facilities; (**c**) the correlation of the settlement and the beach.

Analysis of natural features was the basis for the creation of physical planning documentation in the first observed period [40]. According to the insolation on the terrain (sunrise and sunset, as well as the duration of the terrain exposure to the sun) and the terrain topography (contour line and broken contour line at a right angle), spatial development of the coastal settlement of Prčanj, location Markov Rt, was defined (number 3 in Table 8).

Visual contact, terrain topography and climatic features account for one of the conditions for the spatial development of coastal settlement of Igalo (Herceg Novi municipality) as a health care and tourist center (number 4 in Table 8). The topography of the terrain (Subra massif above the coastal settlement of Herceg Novi, at an altitude of 600–800 m) and the proximity of the coastline, according to General Plan of Boka Kotorska [46], created climatic conditions for the development and planned expansion of the coastal settlement of Igalo in the area of Sutorina (western part of the settlement), all due to the limited space, the use of the coast for health care purposes, the distance from the marina and, consequently, water pollution.

Planned development of the peninsula and coastal settlement of Luštice in the municipality of Herceg Novi is based on natural and anthropogenic features of the area: the only nonbuilt coast of Herceg Novi; the only quality, natural sandy beach in the municipality; a network of settlements at a distance of 1000–1500 m from the coastline, isolated form the neighboring settlement (municipality); the existence of historical properties (fortifications); the harmonization of the transport network with the topography of the terrain; the existence of distinctive elements of the cultivated landscape-olive groves (number 5 in Table 8). The polycentric development of Luštica, based on the existing anthropogenic features, was projected through the functional exchange of nonconcentrated services and activities [34].

The design and position of physical structures in the coastal settlements is greatly influenced by the choice of the location and the traffic–geographic criterion [94]. The need for the construction of a bridge in the Bay of Boka was noted even during the development of the South Adriatic Project [79]. The General Urban Project of Boka Kotorska [34] suggested the position and design of the traffic system between the parts of Herceg Novi municipality, or between Tivat municipality and Herceg Novi municipality, considering the protection of distinctive elements of the natural environment, by building a road on the upper belt and preserving the ambient values of the area (number 9 in Table 8).

The coastal settlement Morinj in the municipality of Kotor has been recognized as a specific residential and tourist point that integrates new, small structures on the slopes with the existing facilities [32]. The proposed modules were defined in a broken form in order to avoid the monotonous appearance of the whole. Fitting the module into the area was the basic idea of the physical planning document. Given the slopes of the area, terrain correction was allowed at a maximum of 1.30 m above the terrain or 1.80 m below the terrain (number 7 in Table 8).

*4.2. Planning Models of the Development of Settlements on the Shores of the Municipalities of Kotor, Tivat and Herceg Novi*

Upon the analysis and valuation of the physical planning documentation, models for the development and improvement of coastal settlements have been identified in this study using previously defined criteria and considering distinctive elements and features of spatial development of coastal settlements. Table 9 shows the proposal for the planned development of the region, municipality or settlement. By obtaining insights into the planning models recognized from the physical planning documentation, it was noticed that no coastal settlement developed on the basis of one model, but on the basis of a combination of two or more of them. For identified models in Table 9, the following acronyms are used: model of the development and improvement of the coastal settlement (preservation of natural values of the identity of coastal settlements through preservation of "OF" and "FROM" views)—"MOF", infrastructural model—IM, economic model—EM, spatial–urban model – SUM and preservation and protection of the historical parts of a settlement-protection and -preservation model—PPM.

**Table 9.** Defining models for further development and improvement based on planning criteria for coastal settlements of Boka, identified through the physical planning documentation.

| | Planning Models | Criteria for the Development of Coastal Settlements and the Existing Natural and Anthropogenic Features | Proposal for the Planned Development | Settlement/Part of a Settlement (Municipality) |
|---|---|---|---|---|
| 1 | MOF; SUM | CNV (respecting terrain topography, insolation, climatic conditions, flora) SUC (reduction of road traffic) | Retaining the basics of spatial organization | Lastva (Tivat) |
| 2 | MOF; IM | CNV (better visual perception of the space, terrain morphology) | Construction of cable cars | Kotor, Muo, Donji Stoliv, Donji Orahovac, Perast (Kotor) |
| | | EC (expanding tourist offer) | | |
| | | CNV (terrain topography, geomechanical characteristics, hydrical features, climatic conditions) | | |
| | | SUC (population, spatial organization) | | |

**Table 9.** *Cont.*

| | Planning Models | Criteria for the Development of Coastal Settlements and the Existing Natural and Anthropogenic Features | Proposal for the Planned Development | Settlement/Part of a Settlement (Municipality) |
|---|---|---|---|---|
| 3 | IM; SUM; PPM | SUC (relieving road traffic) | Sustainable maritime and road traffic | Coastal Region of Montenegro and Boka Kotorska |
| | | CHC (preserving areas of historical significance) | | |
| | | CNV (using natural resources) | | |
| 4 | IM; EM; SUM | CHC (the existence of historical urban ensembles) | Urban patterns of the development of Boka | Boka Kotorska |
| | | CNV (nonbuilt part of a construction area) | | |
| | | EC (developing new tourist centers) | | |
| 5 | SUM; IM | CNV (terrain topography, proximity of the coastline, views) | Spatial organization of the center | Herceg Novi, Topla, Igalo, Savina (Herceg Novi) |
| | | CHC (the existence of a historic core) | | |
| | | SUC (existing urban functions, traffic connections, expanding construction area) | | |
| 6 | SUM; EM; IM | EC (the existence of a large industrial complex, identifying other activities suitable for the area) | Spatial organization of the center | Bijela (Herceg Novi) |
| | | CNV (terrain topography, views) | | |
| | | SUC (promenades, road segregation, ratio of green areas and built-up construction area) | | |
| 7 | SUM; EM; IM | EC (the existence of an industrial plant) | Spatial organization of the center (primary nucleus) | Tivat, Seljanovo (Tivat) |
| | | SUC (uncontrolled expansion of a construction area, linear road traffic, transverse pedestrian promenades, supplementation of public facilities in the coastal zone) | | |
| | | CNV (terrain topography, views) | | |
| 8 | SUM; EM; IM; PPM | CNV (orientation, terrain topography, views) | Spatial organization of the municipality | Kotor, Dobrota, Škaljari, Muo (Kotor) |
| | | CHC (constructions of historical significance, forms of old architecture) | | |
| | | SUC (segregation of busy traffic, defining reserve zones, determining location of tourist and housing capacities) | | |
| | | EC (the existence of an industrial plant) | | |
| 9 | SUM; IM; EM; PPM | CHC (historic core) | Spatial organization of the center | Kotor, Dobrota, Škaljari (Kotor) |
| | | SUC (exemption from construction and road traffic along the coastline, forming a new urban center) | | |
| | | CNV (spatial possibilities of the area, topography of the terrain, views, activation of urban greenery) | | |
| | | EC (the existence of an industrial plant) | | |
| 10 | SUM; IM; EM | CHC (the existence of constructions of historical significance) | Spatial organization of the center | Tivat (Tivat) |
| | | SUC (uncontrolled expansion of a construction area, defining reserve zones, segregation of traffic, strengthening the street front and promenade) | | |
| | | EC (the existence of an industrial plant) | | |
| 11 | PPM; EM | CNV (using topography to organize health care tourism facilities, for agriculture; unsuitable terrain for green areas) | Alternative spatial organizations in the municipality | Igalo, Provodina (Herceg Novi) |
| | | CHC (revitalization of old settlements by conversion into health care tourism) | | |
| | | EC (development thresholds and economic viability of the expansion of the construction area) | | |
| | SUM | SUC (segregation of the construction areas of the coastal settlements) | | |
| | SUM | CNV (using topography for the construction of a marina, views) | | |
| | | SUC (enhancing tourist content, segregation of the functions, construction phases) | | |

By acknowledging the fertility of the land in the fields and along the slopes, urban units of old rural amphitheater-shaped agglomerations were constructed in the history of the origin and development of settlements. They can be used in the further development of settlements as one of the criteria for visual development of coastal settlements. Depending on the area where the structure was built before the observed area, the types of houses can be identified. Respecting the configuration of the terrain, the buildings adapted in terms

of their shape, and thus preserved the natural values of the identity of coastal settlements (number 1 in Table 9).

Making three-dimensional representations of the solution of coastal settlements was an imperative of some physical planning documents from 1968 [95], aiming to identify "focal points" through the subject analysis (architectural research) and examine the connection between the focal points and the natural environment [91]. The main boulevards, which should connect the hinterland with the coastline, also play an important role in defining the visual sequences. The model of development and improvement of coastal settlements through preservation of "OF" and "FROM" views is applied through the spatial–urban and spatial–landscape criteria. After gaining visual perception of the area of municipalities in Boka, the construction of cable cars that would connect the coastal settlement of Boka, i.e., bring the coastline and the immediate hinterland together, was proposed by the General Plan of Boka Kotorska from 1970. The goal of connecting two different belts is the expansion of a tourist offer. In addition, this physical planning document [34] projects adding to the tourist offer by constructing new facilities, hiking trails, restaurants and other content that would activate the locations in questions to a greater extent (number 2 in Table 9).

One of the models of traffic connectivity of coastal settlements proposed by the general Plan of Budva and Ulcinj from 1968 which would result in partial relief of the road traffic, is the inclusion of maritime traffic and minibuses, especially in coastal settlements and areas of historical importance (Bay of Boka Kotorska). This principle would lead to a reduction in the area required for road infrastructure (road parking, bus stops and stations), and would also provide faster traffic and increased attractiveness of the area [34]. A similar principle was proposed by the National Strategy for Integrated Coastal Zone Management of Montenegro and the Coastal Zone Management Program Montenegro (CAMP), but through the implementation of a sustainable mode of transport (vehicles and vessels powered by electricity, solar energy, etc.) [3] (number 3 in Table 9).

Urban development of Boka Kotorska projected by physical planning documentation from the 1970s relied upon the selection of a transport system. Crossing the Bay of Boka Kotorska over or through was planned in two variants: over the bridge or through a tunnel located in Verige or at an alternative location Sveta Nedelja, (Figure 11a) and in Kumbor (Figure 11b). Motorway routes would further affect the centrality of the coastal settlements of Boka (strengthening the coastal settlement of Radovići or Tivat). The emphasized advantage of the urban pattern I refers to the existence of historical urban units which would also contribute to the augmentation of the tourist potential of the area. The advantage of variant II, as explained in the physical planning documentation, is reflected in the possibility of developing the untouched area near the open sea, as well as defining new tourist centers (number 4 in Table 9).

According to the spatial organization of the coastal settlement of Herceg Novi, the area is divided by the traffic network into three zones. Due to weak vertical connections between the functions of the settlement, the spatial–urban model proposes connecting the hinterland with the urban units (port, Old Town, bus station, sports center and buildings with central facilities) as well as with the coastline, defining strong visual directions. Given the terrain configuration, road traffic is defined with side connections and parking lots, without compromising the main view, while pedestrian traffic is projected to continue on the existing one, along the coastline, connecting several coastal settlements into one whole. As proposed, further spatial development of the subject area should be linear, respecting the configuration of the terrain (number 5 in Table 9). The abovementioned physical planning documentation from the first observed period [32] specifies that residential areas, business area, reserve zones, forests, agricultural areas and some parts of the city center should be located in the second zone near the main road. As in the spatial–urban model of Herceg Novi, this model also proposes transverse directions of pedestrian traffic that connect the two zones and at the same time form unimpeded views of the sea (number 6 in Table 9).

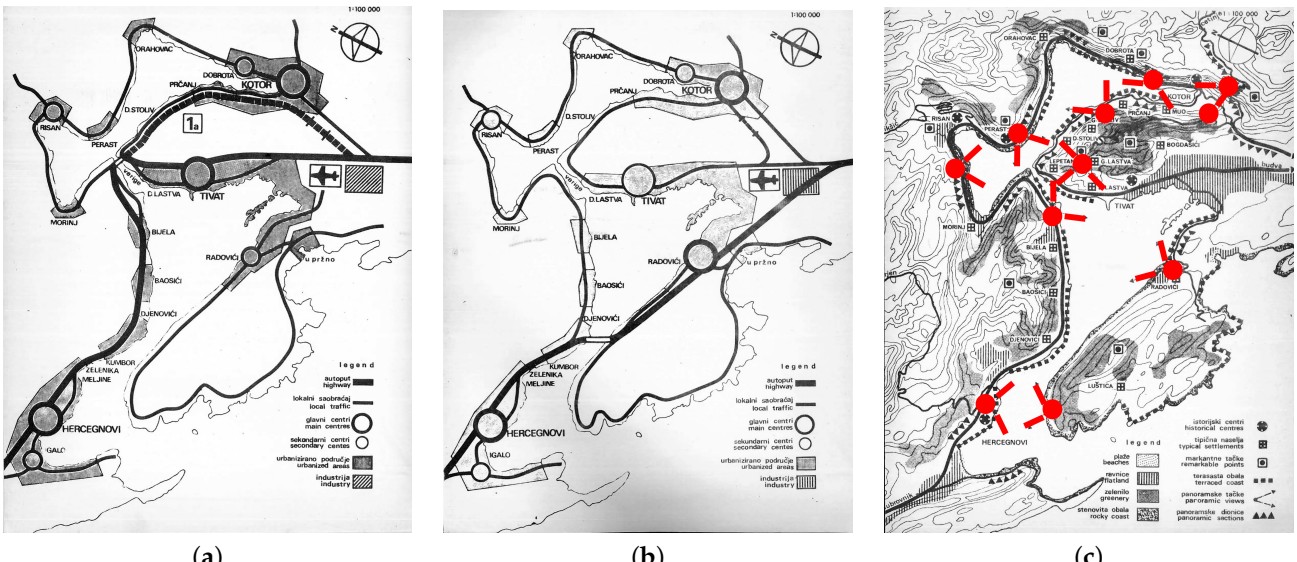

**Figure 11.** Urban pattern of the development of Boka Kotorska: (**a**) Variant I. (**b**) Variant II. (**c**) Analysis of panoramic spots. Panoramic sports are marked red.

Coastal settlement of Tivat was recognized in physical planning documentation from the first observed period [32] as the urban settlement with two nuclei separated by the industrial complex existing at that time. The analysis of the ongoing situation in the 1960s identified the illegal expansion of the construction area of the coastal settlement of Tivat in the area above the main road. Therefore, the subject spatial organization of the settlement projects the expansion of public facilities, i.e., the supplementation of the residential areas with green areas, as well as buildings with educational and central content (number 7 in Table 9). The existing urban functions, such as cultural, administrative and commercial buildings, which are located in the first zone or the zone between the coastline and the main road, are contained in the linear form of the construction area, which is typical for the Mediterranean type of settlements. According to the subject spatial planning document, the public facilities should be supplemented with a school complex adjacent to the city park. The connection of illegally constructed buildings intended for housing in the upper zone and higher housing density in the lower zone is defined through pedestrian promenades that make up for unimpeded views of the waterfront and the sea.

Upon recognizing the limitations or main problems of the area of Boka (uncontrolled development of the construction area of the municipality, lack of urban functions of the settlement, disruption of architectural harmony along the walls of the Old Town, dislocation of industrial complexes, dislocation of roads from the city center), the physical planning documentation from 1970 proposed the following four development zones for Kotor municipality: suppressing and controlling the expansion of the residential area above the main road; supplementing the contents of the Old Town with other urban functions in addition to retaining the residential area; forming a recreational and service area in the coastal settlement of Škaljari; and defining the residential area in the old, attractive fisherman's settlement of Muo (number 8 in Table 9).

Spatial development of the coastal settlement of Kotor and its immediate environment is based on the existence of a historic core functioning as the center of secondary importance due to its spatial potential and limited choice of conversion of structures within the walls of the Old Town. It is proposed to connect the New Town (coastal settlement of Dobrota) and the historic core of Kotor functioning as the secondary center via the marina and the plateau along the coastline in order to ensure the continuation of the tradition of life on the coast [32]. Complementing the contents of the old urban core is projected through the conversion of existing buildings into restaurants, galleries, agencies, shops, etc. The spatial–urban model of the town is focused on relieving the coastline and the entrance of

the Old Town of buildings (banks, courthouse, school, etc.) in order to restore the old image of the historic core and use the free space to compensate for the lack of open public (green) areas within the walls of the settlement (number 9 in Table 9). Residential areas are planned in the neighboring subcenters Dobrota and Škaljari since it is not possible to establish them in the historic core. Both settlements would also include new centers with urban functions that would be in the service of surrounding gravitational areas. Both centers would be located along the coastline, or in its immediate vicinity. The infrastructural model of the city is based on the relocation of the main road behind the walls of the Old Town by building a tunnel, thus relieving the waterfront and the old historic core of the busy traffic, and transforming the coastal road into the main pedestrian link.

In the coastal settlement of Tivat, the spatial–urban model is based on the development of two poles: Tivat (with about 8000 inhabitants) and Seljanovo (with about 5500 inhabitants). According to the physical planning document, the industry is to be dislocated, which would lead to the merging of the two urban units (number 10 in Table 9). The infrastructural model is based on the segregation of the traffic network, with a promenade situated along the coastline, while the two nuclei (urban units) are connected by a secondary road connected to the main road. For both urban units characterized by unplanned construction, in addition to housing, public facilities (center) as well as educational institutions are planned.

While analyzing and determining alternatives for the development of coastal settlements laid in the spatial plan from 1968 [30], the advantages (traffic connectivity, coastline and port formation, terrain topography) and disadvantages (poor coastal belt conditions, pressure of built-up structures of neighboring coastal settlements, the existing built-up land) were identified (Table 10). Based on the aforementioned features, spatial models for further development of coastal settlements of Igalo and Provodine—Herceg Novi municipality, respectively—are proposed (Figure 12, number 11 in Table 9).

**Table 10.** Possible principles of organization of coastal settlements of Igalo and Provodina (Herceg Novi municipality).

| Models | Advantages/Disadvantages |
| --- | --- |
| Model I (Igalo as a multidirectional center): | Advantages: economically favorable expansion of the construction area; health care tourism based on the revitalization of old settlements; use of terrain topography for health care tourism facilities; use of terrain topography for agriculture; use of construction site inconveniences for green areas. |
| Model II (Igalo as the suburb of Herceg Novi): | Advantages: separation of the construction area of coastal settlements of Herceg Novi and Igalo. Disadvantages: construction of Sutorinsko polje (unfavorable for construction); combining the functions of tourism and health care with housing; losing views of the sea. |
| Model III (Igalo as a new tourist center): | Advantages: formation of the marina and use of terrain topography; staged construction; differentiation of functions; expansion of tourist contents; traffic improvement. Disadvantages: specific type of structures and activities. |
| Model II (Modela III variant): | Advantages: as in Model III. Disadvantages: segregation; possible endangerment of the natural environment. |

Planned land reserves and land activation within the municipality may include long-term development zones, (under construction freeze for a period of 15–20 years), medium-term development zones and short-term interventions in space [91]. Certainly,

construction freeze is one of the methods of protecting the land, historical units or parts of settlements from illegal construction, but it also freezes capital.

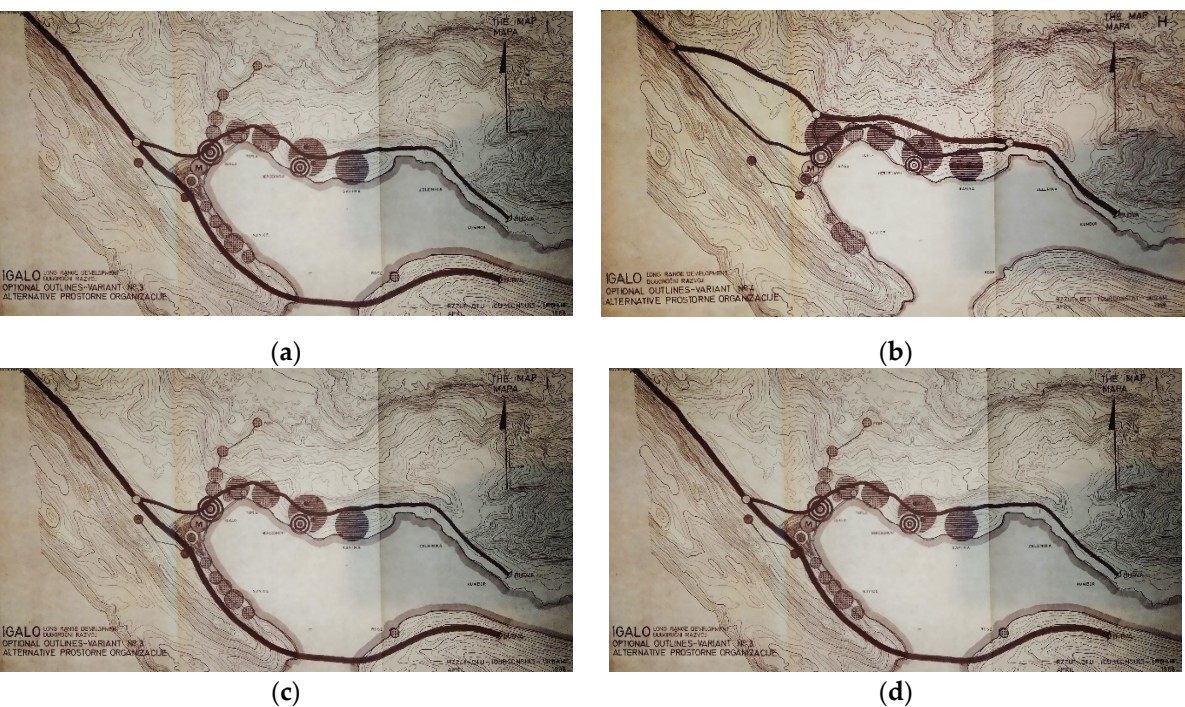

**Figure 12.** Possible principles of the organization of coastal settlements of Igalo and Provodina: (**a**) Model I (Igalo as a multidirectional center); (**b**) Model II (Igalo as the suburb of Herceg Novi); (**c**) Model III (Igalo as a new tourist center); (**d**) Model II (Model III variant).

*4.3. Proposal of New Criteria and Models for the Development of Urban Structures of Coastal Settlements of Boka Kotorska*

Balanced urban development of nonbuilt areas of coastal settlements of Boka is based on the balance between natural features (primarily beach capacities), the areas intended for construction (areas intended for urbanization) and the selection of the areas and buildings for public use. Synchronization of these three criteria for urban development of nonbuilt areas of coastal settlements leads to the formation of a mathematical equation based on the equilibrium of land users' capacities and the area of the land (Table 11).

**Table 11.** Coefficients and capacities of the coast, the coastal zone and the municipality.

| Coefficients/Capacities | | Connection between Natural and Anthropogenic Characteristics |
|---|---|---|
| **Coast coefficients** | Maritimeness | Length of the developed coast (km)/Area of the municipality (km$^2$) |
| | Indentation | Length of the developed coast (km)/Aerial length of the coast (km) |
| | Beach for swimming | Length of coast suitable for swimming (km)/Length of developed coast (km) |
| **Coastal capacities** | Use of the beach for swimming | Length of coast suitable for swimming (m)/Number of tourist beds |
| | Use of the coast | Number of tourists (permanent and excursion)/Length of the developed coast (m) |
| **Coefficient of littotality** | | Area of the coastal zone (500 m) (km$^2$)/Area of the municipality (m$^2$) |

Planning (spatial–urban) criteria for the development of coastal settlements must depend on the capacity of the beach, i.e., natural and demographic characteristics (beach area, length of the coastline, number of tourists and number of permanent residents), or on the coefficient of the use of the swimming beach, swimming coast coefficient and the beach coefficient. Moreover, all three said criteria should be harmonized with the littotality coefficient, i.e., with the spatial capacities of a coastal settlement in relation to a coastal municipality. This coefficient will furthermore affect the dispersion of the construction area of coastal settlements within a municipality (Figure 13).

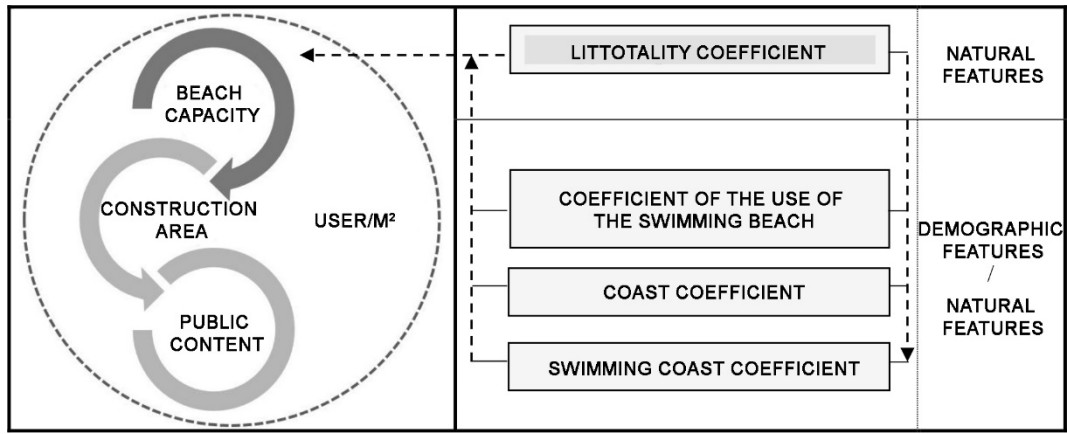

**Figure 13.** Planned criteria for the development and improvement of coastal settlements.

If the built-up area and, consequently, the number of users exceed beach capacities of a coastal settlement and the tourist standard decreases, this type of a settlement can only be revitalized, but not expanded, under physical planning documentation. To compensate the lack of space on these beaches, the areas of public facilities should be used. In case of settlements characterized by the existing balance between the beach capacity and the construction area (i.e., the number of space users), and the lack of public facilities, it is possible to expand the settlement or provide more said facilities, considering the ratio of the beach area and the number or inhabitants/tourists.

Natural values criteria of coastal settlements of Boka can be evaluated using the coast indentation index that affects shaping the construction areas of a coastal settlement. The closer the index is to 1.00, the straighter the coastline, and vice versa (Table 12). This concept is closely related to views that represent the basic element of the model of development and improvement of coastal settlements or model of the preservation of natural values of the identity of coastal settlements through preservation of "OF" and "FROM" views. A coast that is more uneven has more natural locations with focal points (capes). In coastal settlements where the coastline is straighter, the spatial–urban model can artificially define greater dynamism of the coast with new focal points on the coastline (marinas, quays, mandrakes, anchorages, ports, etc.). The consequence of the artificial design of the coastline affects the traffic system (maritime, pedestrian and road traffic).

**Table 12.** Coast indentation index as a planned criterion for the development of a coastal settlement.

| | Coast Indentation Index (If) | |
|---|---|---|
| **Influence on a Coastal Settlement** | **If = → 1** (Closer to the Value of 1.00) | **If = ← 1** (Further from the Value of 1.00) |
| Traffic system: | Road traffic dislocation; segregation of road traffic from pedestrian traffic; shaping roads along the coastline. | Possibility of cross-linking the coastline and the construction area through pedestrian traffic. |
| Visual perception: | Smaller angle of the visual field; forming artificial structures on the coastline. | Greater potential for using the natural dynamics of the coastline. |

In addition, the abovementioned index is closely related to the infrastructural model. The greater the indentation of the coast, the more dislocated road traffic should be from the coastline. This approach frees the narrow coastal belt (the belt up to 100 m from the coastline) from the road traffic and enables the construction of a promenade along the coast. As described in the previously analyzed physical planning documentation, the longer the connection of coastal settlements with the promenade, the stronger the effect of the spatial–urban model of the coastal municipality (Figure 14).

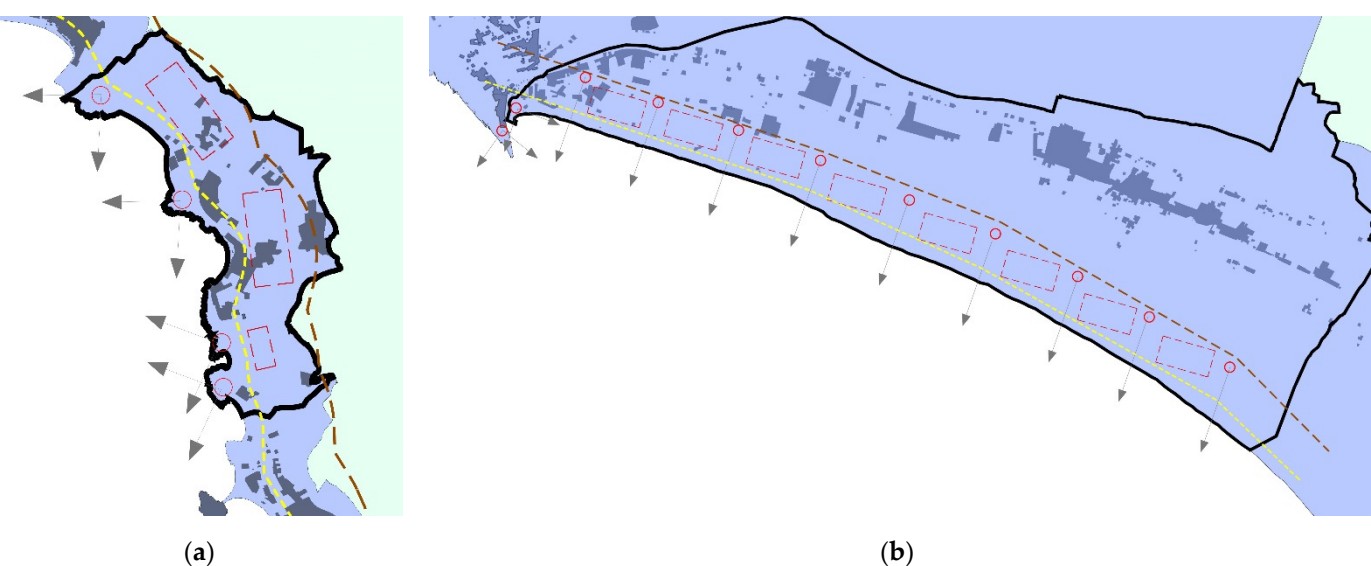

|   |   |
|---|---|
| (**a**) | (**b**) |

**Figure 14.** An example of the planned development of a coastal settlement depending on the coefficient of indentation of the coast; (**a**) coastal settlement Pržno (Budva); (**b**) Donji Štoj (Ulcinj).

This coefficient also affects the possibility of expansion of the building part of the settlement (indicated by red rectangles in Figure 14), and the formation of artificial structures and shaping of natural elements (corners of the field of vision and horizontal limits of vision marked by red circles and dashed black lines in Figure 14).

In coastal settlements of Boka, industrial buildings of historical importance were not listed as cultural heritage sites. In the area of the coastal settlements of Tivat, Kotor, Risan, Bijela and Kumbor, there are several undeveloped locations of once-active industrial complexes that had historical significance. These facilities can also be part of the model of preservation and protection of historical parts of the settlement. Construction areas of coastal settlements should be separated from historic cores and units by means of open green areas or green breaks, which would serve to stop the construction and preserve the coastline. This would help to avoid scattering along the coastline and the formation of conurbations of settlements, as well as linear stretching of the built-up part of the settlement along the coast. This model of the preservation of natural values of the identity of coastal settlements through preservation of "OF" and "FROM" views is closely related to the infrastructural model.

## 5. Conclusions

This paper deals with the research of the spatial development of Montenegrin settlements on the Adriatic coast through their planning framework, as well as proposals for possible guidelines for their further development and improvement. However, the procedures for analysis and evaluation of certain data, and the procedure for obtaining results, can be applicable to other coastal settlements in the world. By comparing the researched and known criteria for the development of settlements (landscape, cultural–historical, spatial–urbanistic, economic and legislative), this paper determined the general and/or individual criteria for the development and improvement of these settlements with the aim of establishing clear, general and individual models for their future planning development.

Data on the construction of the coastal settlements of Boka Kotorska conducted within the time frame from 1945 to 1992, and in the period around the year of obtaining independence, clearly show that the expansion and pressure on the construction subject area followed after 1992. The consequences of urban transformations are best reflected in the coastal setback (a zone of 100 m from the coastline to the mainland), which has become the most vulnerable area of the subject area. Urban transformations of the subject area have also been analyzed from the perspective of the position and expansion of the built-up

land at different altitudes, i.e., above and below 100 m above sea level. The results indicate that the sudden changes that occurred in the Bay of Boka Kotorska after 1992, amid great pressure to expand the construction area, often neglected the natural features and spatial possibilities of planned development of coastal settlements.

A detailed analysis of the research on the relationship between space, features, criteria and models has identified those criteria and models that can be applied to both nonbuilt and/or ruined parts of the settlement, which are recorded through physical planning documentation, as well as to new interventions with similar natural and anthropogenic features. The examination of natural features of the subject area has revealed certain criteria for natural values of the coastal settlements.

In order to investigate the changes in the area of coastal settlements, the identity features of the natural (geomechanical, pedological features, features of the relief and other phenomena in the area, as well as flora and fauna) and the built environment (characteristics of the cultivated, historical–cultural environment, shape, the size and gravitational force of a building area and finally the economic characteristics and traffic network) were examined. The analysis of the abovementioned identity features, recognized through the PPD, confirmed the importance of the assumed features and their influence on the spatial and planning development of the settlement: the topography of the land on which the coastal settlement was built, the cultural assets protected by law and the identity features of the built environment (traffic network and identity features of the cultivated environment).

In most PPDs, there is no scientific approach to recognizing and connecting identity features or their impact on the value of coastal settlements. The same applies to the criteria used to describe their impact on space. Accordingly, in this work, by analyzing the identity features from the natural and anthropogenic (cultivated, cultural and built) environment, criteria were determined according to which the mentioned features can propose guidelines for the planned development of the area. By recognizing the newly established planning criteria, in this work, planning models were consequently determined which can be of great importance for the development of coastal settlements.

Based on the conducted research, it is necessary to add the following criteria, capacity of coastal settlements, which can be determined based on the coefficients and capacity of the coast, the coastal zone and the municipality. The use and further analysis thereof would have a major impact on the design of the construction area of Boka. The proposed planning criteria (coefficients) have not been recognized as mandatory at the practical level. By applying the recognized subject coefficients, which are applicable to coastal settlements and municipalities, planning models that can be used in the preparation of physical planning documentation can be defined.

Defining coastal setback in the longitudinal direction represents a part of the process of compiling physical planning documentation of the coastal region of Montenegro [28], i.e., one of the criteria of the spatial–urban model of development of coastal settlements of Boka Kotorska. The proposal of mandatory definition of coastal setback through the application of certain indicators of spatial development should be made on the basis of input data, i.e., on the basis of analysis of consistent criteria for further urban development of coastal settlements. By analyzing data using modern methods (e.g., GIS), we can obtain a quality database that would be used to monitor the situation in the field and help define the most straightforward methodology for the development of physical planning documents. Scientific research on the development of coastal settlements in Europe in the period 1990–2005 resulted in defining certain criteria or indicators of the state of the coastal zone environment [1]. The goal of further improvement and development of coastal settlements is to examine the subject indicators over time in order to make the comparison of data more efficient and easier.

**Author Contributions:** Conceptualization, S.S.; methodology, S.S.; formal analysis, S.S. and N.M.; resources, S.S.; writing—original draft preparation, S.S.; writing—review and editing, S.S., J.B.Š. and N.M.; visualization, S.S. and J.B.Š.; supervision, J.B.Š. and N.M. All authors have read and agreed to the published version of the manuscript.

**Funding:** This research received no external funding.

**Institutional Review Board Statement:** Not applicable.

**Informed Consent Statement:** Not applicable.

**Data Availability Statement:** The data presented in this study are available on request from the corresponding author. The data are not publicly available due to the legal status of the company that keeps the archival documentation used in the preparation of this paper.

**Acknowledgments:** Institutions that store archival documentation used in the preparation of this paper, and to whom we are grateful for providing us with the material of spatial planning content, are: The Republic Institute for Urbanism and Design a.d.d. Podgorica, the Secretariat for Urban Planning, Construction and Spatial Planning of Kotor Municipality and the Cadaster and State Property Administration of Montenegro.

**Conflicts of Interest:** The authors declare no conflict of interest.

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
