# Peer review of "Planning Criteria and Models for the Development of Urban Structures of Coastal Settlements of Boka Kotorska"

_sustainability, doi:10.3390/su14159467_

Round 1

Reviewer 1 Report

This paper examines planning criteria and models for the development of urban settlements situated in Montenegro, Boka Kotorska during the period between 1945 and 2006. Paper  has good structure. In order to be more readable the authors should consider the following:

Figure 2 contains labels on Montenegrin language. Also, I'm not sure if such image is necessary. It takes lots of space.

Also Table 1 is not necessary since the explanation of the terms are already given in text.

Table 2 should be placed right after it is first mentioned.

Figure 3. Description of what the colors on maps represent should be placed in the text

Authors should explain which tools did they used for all the results, which type of GIS did they used, what is the quality of the data they used, did the data on cadastral parcel came form real estate cadastre with the timestamp, or what was the origin of the data from different time periods.

Did authors analysed the data about demolished buildings and how that changes the results.

Reviewer 2 Report

This is a meaningful and well-written article. It is suggested to be accepted after minor revision. A few comments for reference:

       (1) Many pictures in the paper are not clear, so the resolution needs to be improved.

       (2) Key scientific questions and marginal contributions to be addressed by research need to be further clarified.

       (3) The reason for choosing the time period of the study needs further explanation, because the author has not updated the latest time.

Reviewer 3 Report

The work aims to examine planning criteria and models for the development of urban settlements situated along the coastline of Boka Kotorska. Some improvements are required in order to publish the work:

1) Abstract

Please, provide for clearly stating the methodology used in the work and the innovative contributions and also the usefulness in the practice; 

2) Introduction

Please, insert some international references on the matter of the urban development and its consequences. Some could be: Morano, P., Tajani, F., & Anelli, D. (2021). Urban planning variants: A model for the division of the activated “plusvalue” between public and private subjects. Valori e Valutazioni, (28) and Camagni, R. (2017). Urban development and control on urban land rents. In Seminal Studies in Regional and Urban Economics (pp. 283-302). Springer, Cham.

3) Materials and methods

Please, provide clear expaination of the indentation index for the aim of the work

4) add a flowchart of the analized document with a brief description of it

Reviewer 4 Report

The paper examines planning criteria for supporting the development of urban settlements located along the coastline in a specific area of Montenegro.

The topic of coastline development and how to support the definition of policies and strategies for their planning and safeguarding could be potentially interesting and relevant. However, a strong revision is needed before possible publication in an international journal. The paper, in fact, simply focuses on Montenegro, without providing a common methodology that other scholars could apply to other contexts. It is also testified by a complete absence of an analysis of the international literature on this topic. The paper only uses references from Montenegro.

Moreover, the paper is not well structured. It makes the reading quite difficult!

A strong revision of the paper is therefore needed. I listed below the main reviews to be made for each section:

INTRODUCTION:

1.      The proposal must be included in the current international research debate and justified through it.

2.      Similarly, the innovation of the research must be stated according to the current international debate. How does your proposal provide additional knowledge or an alternative methodology to study that topic?

3.      Most of the details about Montenegro should be moved to the case study description (Section 2.1)

4.      The aim and scope of the research cannot be simply limited to Montenegro: you can provide a common framework/scheme/methodology to be applied to other case studies worldwide.

5.      The last part of the introduction should have a brief description of the contents of each section.

MATERIAL AND METHODS:

6.      Only Section 2.1 must contain information about Montenegro and the case study area. Move here (Section 2.1) the information contained in the introduction and in the following sections.  

7.      Moreover, the case study should be represented in the wider UE context, not simply an image (Fig 1) with the detailed subdivision of settlements of coastal municipalities.

8.      According to comment 6, the other sections of Material and Methods should contain the methodology adopted. The research methodology proposed must be presented here in order to provide a common framework/scheme for scholars to be applied to other case studies worldwide.

9.      Only after the description of the general framework/methodology, you can detail the data that you have used for the case study of Montenegro.

10.   It is not very clear to me how you move from the analysis of laws to the definition of the features analyzed and then the criteria. Please, describe very well the process in this section (Material and Methods), otherwise, the following sections are hard to understand.

11.   It is not very clear why you stopped your analysis in 2006. It is quite strange, so please justify it.

12.   In fig 2, show the subdivision of the two periods considered (1945-1992 and 1992-2006)

13.   In tab 2, it is not clear the difference between “1” and “X”. Please, describe it better.

14.   In tab 2, use an acronym also for the laws

15.   I am not sure that an in-depth description of all Montenegro laws (Section 2.2.) is needed for international scholars. You can simply provide a unique table (starting from Tab 2) giving a brief description of each law (you can use an entire horizontal page for the table).

16.   Then, you can use tab 2 to clarify the features analyzed and then the selection of the set of criteria. In particular, how do you define it? According to what? Availability of data, Literature review, Laws, Territory characteristics? An in-depth description of the process that you follow for defining the criteria is fundamental!!!

17.   You can use an image that describes the entire process that you have followed.

RESULTS:

18.   The Result Section is too long and not well structured. I suggest to include a DISCUSSION Section in the paper, in order to split the contents of Section 3. In particular, you can move Section 3.3 and/or 3.4 in the Discussion Section

19.   Tab 4 and 5 should be merged, otherwise, it is hard to understand the implications and information of tab 5.

20.   Since you analyzed the changes in the built-up land in the two periods, I expected an analysis of the relation between built-up land and population. However, the analysis of the population was in another section, and it considered a different timeframe. Please, provide more information.

21.   Is Tab 7 needed at the end of Section 3.2 or can be anticipated?

22.   Without a previous description of the elements considered (in Material and Methods), Section 3.2.3 is hard to read.

CONCLUSIONS:

23.   Conclusions need to recall the aim and scope of the research, and then describe the main results and findings.

24.   Again, the findings of the research should be considered 

Round 2

Reviewer 1 Report

Authors corrected all remarks and the paper is better now, so I suggest to editor to accpet the paper for publication.

Reviewer 3 Report

The efforts made by the Authors are apprecciated.

Reviewer 4 Report

There are some problems in the pdf in line 304, as well as in the references (e.g. 2-8, 14-24)

English language and style are fine/minor spell check required